**RESEARCH**                                                                                          **Open Access**

# DNA methylation repels binding of hypoxia-inducible transcription factors to maintain tumor immunotolerance

Flora D'Anna[1,2†], Laurien Van Dyck[1,2†], Jieyi Xiong[1,2†], Hui Zhao[1,2†], Rebecca V. Berrens[3,4], Junbin Qian[1,2], Pawel Bieniasz-Krzywiec[1,5], Vikas Chandra[6], Luc Schoonjans[1,7,8], Jason Matthews[9], Julie De Smedt[10], Liesbeth Minnoye[1,2], Ricardo Amorim[1,5], Sepideh Khorasanizadeh[6], Qian Yu[11], Liyun Zhao[11], Marie De Borre[11], Savvas N. Savvides[12,13], M. Celeste Simon[14,15], Peter Carmeliet[1,7,8], Wolf Reik[3,16,17,18], Fraydoon Rastinejad[6,19], Massimiliano Mazzone[1,5], Bernard Thienpont[1,2,11*] and Diether Lambrechts[1,2*]

* Correspondence: bernard.
thienpont@kuleuven.be; diether.
lambrechts@vib-kuleuven.be
†Flora D'Anna, Laurien Van Dyck,
Jieyi Xiong and Hui Zhao
contributed equally to this work.
[1]Center for Cancer Biology, VIB,
3000 Leuven, Belgium
Full list of author information is
available at the end of the article

## Abstract

**Background:** Hypoxia is pervasive in cancer and other diseases. Cells sense and adapt to hypoxia by activating hypoxia-inducible transcription factors (HIFs), but it is still an outstanding question why cell types differ in their transcriptional response to hypoxia.

**Results:** We report that HIFs fail to bind CpG dinucleotides that are methylated in their consensus binding sequence, both in in vitro biochemical binding assays and in vivo studies of differentially methylated isogenic cell lines. Based on in silico structural modeling, we show that 5-methylcytosine indeed causes steric hindrance in the HIF binding pocket. A model wherein cell-type-specific methylation landscapes, as laid down by the differential expression and binding of other transcription factors under normoxia, control cell-type-specific hypoxia responses is observed. We also discover ectopic HIF binding sites in repeat regions which are normally methylated. Genetic and pharmacological DNA demethylation, but also cancer-associated DNA hypomethylation, expose these binding sites, inducing HIF-dependent expression of cryptic transcripts. In line with such cryptic transcripts being more prone to cause double-stranded RNA and viral mimicry, we observe low DNA methylation and high cryptic transcript expression in tumors with high immune checkpoint expression, but not in tumors with low immune checkpoint expression, where they would compromise tumor immunotolerance. In a low-immunogenic tumor model, DNA demethylation upregulates cryptic transcript expression in a HIF-dependent manner, causing immune activation and reducing tumor growth.

**Conclusions:** Our data elucidate the mechanism underlying cell-type-specific responses to hypoxia and suggest DNA methylation and hypoxia to underlie tumor immunotolerance.

**Keywords:** DNA methylation, Hypoxia, HIF, Cryptic transcripts, Immunotherapy, Cancer, Transcription factor binding

## Background

DNA methylation is central to establishing and maintaining tissue-specific gene expression, and an important contributing factor to oncogenesis. We recently demonstrated that pervasive and ablating conditions of tumor hypoxia drive DNA methylation of tumor suppressor genes by reducing the activity of TET DNA demethylases [1]. An outstanding question is, however, if and how DNA methylation in turn also influences the response of tumors to (acute) hypoxia. Indeed, recent evidence suggests that, contrary to traditional concepts, DNA methylation generally does not directly impede transcription factor (TF) binding, but rather acts indirectly by synergizing with other epigenetic marks [2].

The hypoxia response is canonically executed by HIFs, which are heterodimeric TF complexes composed of an $O_2$-labile α-subunit (HIF1α, HIF2α or HIF3α) and a stable β-subunit (HIF1β). The constitutively expressed HIFα subunits are directly targeted for proteasomal degradation under normal oxygen tension (normoxia), but stabilized under limiting oxygen conditions (hypoxia), when they translocate to the nucleus to induce expression of hypoxia-responsive genes. This induction of hypoxia-responsive genes occurs rapidly, often within minutes following hypoxia [3]. In tumors, hypoxia is widespread and leads to transcriptional activation of numerous cancer hallmark genes involved in cell survival, angiogenesis, and invasion [4]. Interestingly, the impact of hypoxia differs among cell types. For instance, endothelial cells proliferate and migrate towards hypoxic regions, macrophages become immunosuppressive and CD8[+] T cell activation is enhanced under hypoxia [5–8]. Also, tumors affecting different organs exhibit divergent phenotypic responses to hypoxia [6]. In line with this, concordance between HIF binding sites in MCF7 breast and 786-O renal cell carcinoma cell lines is only 40–60% [9]. This divergence is particularly intriguing because HIFα paralogues are often expressed at similar levels in different cell types, and because the consensus DNA sequence that binds HIF complexes, i.e., the hypoxia response element (HRE) RCGTG, does not differ between HIFα paralogues or cell types. Thus, although the concept that genes induced by hypoxia differ dramatically between cancer and cell types is well-established [3, 10–12], the reason for these divergent responses and expression programs is poorly understood.

One possibility is that the underlying cell-type-specific patterns of chromatin determine which HIF target genes are accessible and hence become expressed following acute hypoxia. Interestingly, HIFs are recruited to genes that are already expressed in normoxic cells [12], suggesting that perhaps DNA methylation could determine accessibility of the HIF complex to the RCGTG core sequence. HIF binding to the erythropoietin promoter was indeed suggested to be sensitive to DNA methylation [13], but this observation relied on gel shift binding assays, which are known to poorly reflect the authentic setting in cells. Indeed, the binding of the transcriptional repressor CTCF also appeared to be methylation-sensitive in gel shift binding assays [14], but analyses of its binding preference in living cells mostly failed to reveal methylation sensitivity [15]. We therefore set out to investigate whether DNA methylation directly repels HIF binding in living cells, and whether cell-type-specific DNA methylation patterns established under normoxic conditions determine genome-wide HIF binding profiles, defining the response to hypoxia.

## Results

### DNA methylation of HRE sites anti-correlates with HIF binding

To investigate the role of DNA methylation in HIF binding, we stabilized HIFs in MCF7 breast cancer cells by culturing them under acute hypoxia (0.5% $O_2$ for 16 h; Additional file 1: Fig. S1a and S2, and "Methods"), conditions that are insufficient to drive hypoxia-induced hypermethylation [1]. We next performed chromatin-immunoprecipitation coupled to high-throughput sequencing (ChIP-seq) for HIF1β, which is the obligate dimerization partner of HIF1α, HIF2α, and HIF3α. Model-based analysis for ChIP-seq (MACS) [16] revealed 7153 HIF1β binding peaks (Fig. 1a, Table S1). These were high-quality, bona fide HIF binding regions: they were 4.6-fold enriched for the HRE motif (RCGTG), enriched near genes involved in the hypoxia response, > 90% overlapping with peaks identified in another HIF1β ChIP-seq dataset on MCF7 cells and reproducibly detected in independent repeats (Additional file 1: Fig. S1b-d).

To assess methylation in these 7153 HIF1β binding peaks, we performed target enrichment-based bisulfite sequencing (BS-seq) on DNA extracted from normoxic MCF7 cells, in which HIF is inactive, obtaining > 40× coverage for ~ 86% of the HIF1β binding peaks identified by ChIP-seq. The methylation level at these peaks was invariably low (4.95 ± 0.15%) compared to average CpG methylation levels detected in the genome (61.6 ± 0.07%, Wilcoxon test $P < 2.2^{-16}$, Fig. 1b). Results were confirmed using another whole-genome BS-seq dataset (Fig. 1a) [18]. Also when quantifying methylation across all RCGTG motifs, including those located outside of HIF1β binding peaks, the inverse correlation between DNA methylation and HIF binding was confirmed (Fig. 1c). As BS-seq does not discriminate between 5-methyl (5mC) and 5-hydroxymethylcytosine [19], we confirmed by DNA immunoprecipitation with an antibody recognizing only 5mC (5mC-DIP-seq) that HIF1β binding peaks were six fold depleted in 5mC-DIP-seq reads (Fig. 1a). Moreover, methylation analysis of normoxic *HIF1B*-knockout MCF7 cells [20] revealed identical methylation patterns (Additional file 1: Fig. S1e-g), indicating that the unmethylated state of HIF1β binding sites is not due to baseline activities of HIF1β under normoxia. Importantly, identical results were obtained for murine embryonic stem cells (mESCs): the loci corresponding to the 4794 HIF1β binding sites identified in wild-type ESCs were unmethylated in normoxia, and this both in wild-type and *Hif1b*-knockout ESCs [21] (Additional file 1: Fig. S1h-j). Since cells were intentionally exposed only briefly to hypoxia (16 h), which fails to induce pronounced DNA methylation changes [1], these data suggest that regions to which HIF1β binds upon hypoxia are devoid of DNA methylation under normoxic conditions.

### Cell-type-specific DNA methylation of HREs determines HIF binding

Different cell types respond differently to hypoxia. To assess whether cell-type-specific DNA methylation could underlie this phenomenon, we profiled DNA methylation and HIF1β binding in 2 additional cell lines (RCC4 and SK-MEL-28). A total of 20,613 HIF1β binding peak positions were detected across these cell lines (Additional file 3). For each cell line, HIF1β binding was annotated as "present" if the peak area showed > 4-fold enrichment over the local read depth, and as "absent" if it showed < 2.5-fold enrichment; intermediate enrichment was annotated as unclassified (Additional file 3). When comparing cell lines using these criteria, HIF1β binding was shared by all 3 cell lines at 6152 sites,

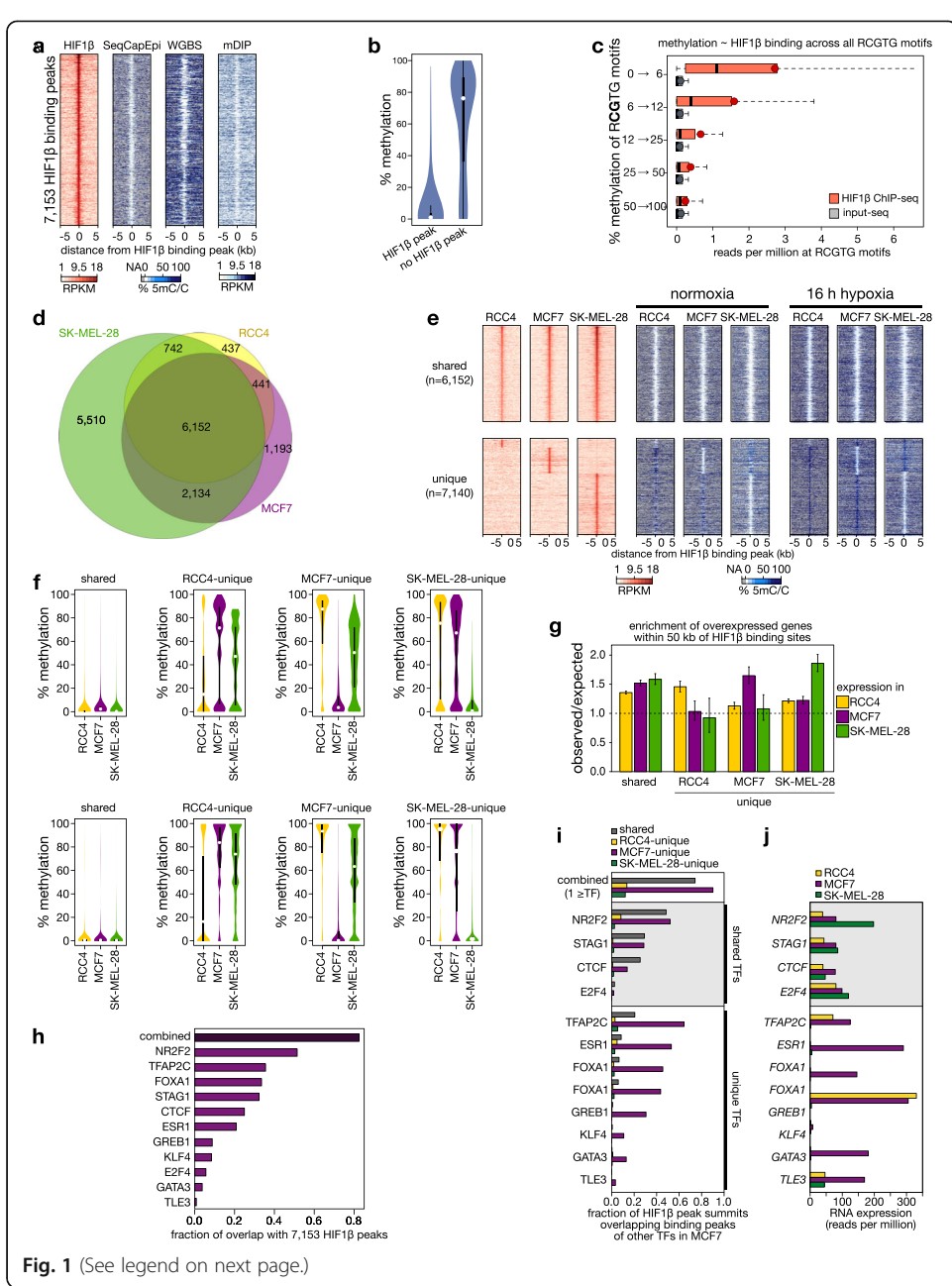

**Fig. 1** (See legend on next page.)

(See figure on previous page.)

**Fig. 1** Methylation at HIF1β binding sites. **a** Heatmaps of HIF1β binding and DNA methylation for 7153 regions (identified using a stringent threshold of $P < 10^{-15}$ in MACS) surrounding the HIF1β ChIP-seq peak summit (± 5 kb). Heatmaps depict reads per kb per million reads (RPKM) of HIF1β ChIP-seq and of 5mC DNA IP-seq (mDIP), and % DNA methylation as estimated by SeqCapEpi BS-seq or whole-genome BS-seq (respectively, SeqCapEpi and WGBS). HIF1β binding was assessed after 16 h of 0.5% $O_2$ (hypoxia) and DNA methylation under 21% $O_2$ (normoxia). **b** Violin plots of the methylation level inside and outside HIF1β binding peaks, as estimated by SeqCapEpi BS-seq. **c** Sequencing read depth of HIF1 ChIP and its input, at all RCGTG sequences in MCF7 cells, stratified for methylation at the CG in the core RCGTG sequence. Shown are boxplots for all RCGTG's in the human genome for which > 10× coverage was obtained after SeqCapEpi BS-seq, with dark red dots denoting averages. See Additional file 1: Fig. S5 for additional QC of ChIP-seq data and Additional file 2 for more details about HIF1β binding peak locations. **d** Venn diagram of 20,613 shared and unique HIF1β binding sites detected across 3 cell lines. Only stringent binding sites ($P < 10^{-15}$) are shown. Binding sites showing intermediate levels of HIF1β ChIP-seq enrichment in 1 or 2 cell lines are unclassified and not shown here ($n = 445$, 2812 and 887 peaks, detected in SK-MEL-28, RCC4, and MCF7 respectively). **e** Heatmaps of HIF1β binding (*red*) and DNA methylation as estimated using SeqCapEpi BS-seq (*blue*) at regions flanking the HIF1β ChIP-seq peak summit (± 5 kb). (*top*) HIF1β binding peaks shared between the 3 cell lines. (*bottom*) HIF1β binding peaks unique to each cell line. Heatmaps depict RPKM of HIF1β ChIP-seq and % DNA methylation. HIF1β binding was assessed after 16 h of 0.5% $O_2$ (hypoxia) and DNA methylation under 21% $O_2$ (normoxia) or after 16 h of 0.5% $O_2$ (hypoxia). **f** Quantification of the methylation level at HIF1β binding peak summits ± 100 bps, for peaks that are shared between or unique to one of the 3 cell lines grown under 21% $O_2$ (normoxia, *top*) or after 16 h of 0.5% $O_2$ (hypoxia, *bottom*). **g** Enrichment of gene expression (observed/expected) upon hypoxia per cell line, for genes associated with HIF1β binding sites (within 50 kb) that are shared between or unique to one of the three cell lines, as labeled on the *X*-axis. **h** Fraction of HIF1β peaks overlapping with the binding peaks of individual transcription factors [17], or with any of the 11 transcription factors profiled in MCF7 cells ("*combined*"). **i** Overlap between HIF1β binding peaks and other transcription factor binding sites detected in MCF7 cells. Shown are fractions of HIF1β binding peaks shared between (gray) or unique for a cell line (colored). **j** mRNA expression level of transcription factors in each cell line, as determined using RNA-seq. Transcription factors expressed in all three cell lines are highlighted as "*shared TFs*" with a light gray box

and unique for an individual cell line at 7140 sites (437, 1193, and 5510 unique sites, respectively for RCC4, MCF7, and SK-MEL-28) (Fig. 1d, Additional file 1: Fig. S1k-l). Crucially, when assessing DNA methylation both under normoxia and under acute hypoxia, HIF1β binding peaks unique to individual cell lines were unmethylated in cells where the binding site was active, while active HIF1β binding peaks shared between all cell lines were unmethylated in all cell lines (Fig. 1e, f, Additional file 1: Fig. S1m-n). This strict correlation suggests that DNA methylation underlies the cell-type-specific response to hypoxia. Differences in DNA methylation and concomitant HIF binding also appeared functional, as transcriptome profiling under normoxic and hypoxic conditions revealed that genes with a flanking HIF1β binding peak unique to one cell line were more frequently increased in expression under hypoxia in that cell line (Fig. 1g).

### DNA methylation determines HIF binding independently of other chromatin marks

To analyze whether other epigenetic modifications similarly correlate with HIF binding, we analyzed public ENCODE data for MCF7 cells [22] (no data are available for RCC4 and SK-MEL-28). Particularly, we investigated marks of heterochromatin (H3K9me3, H3K27me3), active promoters (H3K4me3, H3K9ac, H3K14ac), active enhancers (H3K4me1, H3K27ac), open chromatin (FAIRE), and active transcription (RNA Pol-II). Although some histone marks were enriched in a subset of HIF1β binding peaks, none were consistently found at all active HIF1β binding peaks, especially when looking outside of CpG islands (Additional file 1: Fig. S3a). The previously reported co-occupancy with RNA polymerase II or open chromatin was also not consistently found at all active

HIF1β binding peaks [12]. This was confirmed in linear regression analyses assessing how each mark individually predicts HIF1β binding in MCF7 cells. DNA methylation ($R^2 = 0.43$) outperformed all other marks, with marks of active chromatin such as RNA polymerase II occupancy, H3K4me3, open chromatin, and H3K27ac showing poor correlations ($R^2$ resp. 0.11, 0.11, 0.04 and 0.04). When combining all marks in one model, the total $R^2$ was 0.47, with DNA methylation contributing to 67.5% of the predictive power (partial $R^2 = 0.32$). In line with this, omitting DNA methylation from the model reduced the total $R^2$ by more than half, to 0.21 (Additional file 1: Fig. S3b).

We also assessed whether more general differences in chromatin states (using ChromHMM [23]) underlie differential HIF binding. This revealed that while shared HIF1β binding sites were more frequent in promoters, sites unique to MCF7 were more frequent in enhancers, and sites inactive in MCF7 (but unique to RCC4 or SK-MEL-28) more frequent in MCF7-repressed chromatin (Additional file 1: Fig. S3c). In line with enrichment at open chromatin, HIF1β binding thus appears exclusive to active enhancers and promoters while depleted in areas of repressed chromatin. Finally, NOMe-seq data from MCF7 cells revealed that, while open chromatin regions were generally unmethylated, a significant fraction of open chromatin (7–19%) in fact showed methylation (Additional file 1: Fig. S3d), providing a potential rationale for the relatively small contribution of open chromatin to predict HIF1β binding. Combined, these data show that in normoxia poised HIF binding sites are located in unmethylated regions that consist mostly of active, open chromatin but are not consistently marked by other epigenetic modifications.

### Other TFs determine the methylation landscape to guide HIF binding

Interestingly, many of the HIF1β binding peaks overlapped with binding sites for other TFs. Specifically, out of the 7153 HIF1β binding peaks detected in MCF7 cells, 5903 overlapped with the binding site of at least one TF (83%), out of a set of 11 TFs for which genome-wide binding site data were available in MCF7 cells [17] (Fig. 1h). This could indicate that these TFs, being already active under normoxic conditions, drive demethylation of HIF1β binding regions [24], thus setting the stage for HIF binding upon hypoxia. To further support this notion, we assessed whether these 11 TFs also bind at HIF1β binding peaks identified in RCC4 and SK-MEL-28 cells. Interestingly, TFs expressed by the 3 cell lines (e.g., CTCF or STAG1) co-localize in their binding with the shared HIF1β binding peaks. In contrast, TFs only expressed in MCF7 cells (e.g., ESR1 or GATA3) overlap in their binding sites only with MCF7-specific HIF1β binding peaks. Finally, binding of these 11 TFs in MCF7 did not overlap with HIF1β binding peaks unique to RCC4 or SK-MEL-28 (Fig. 1i, j). These data were replicated in an independent cell line (Additional file 1: Fig. S3e-g). Differential expression and binding of TFs between different cells is thus likely to shape the DNA methylation landscape and determine subsequent HIF binding.

### DNA methylation does not determine differential binding of HIF1α and HIF2α

Comparison of our 7153 HIF1β peaks to previously published HIF1α and HIF2α ChIP-seq data in MCF7 cells revealed that the methylation status of HIF1β binding peaks was independent of the HIFα binding partner, as HIF1α- and HIF2α-bound DNA showed similar methylation levels (Additional file 1: Fig. S3h). Remarkably, there were

differences in the chromatin profiles of HIF1α- and HIF2α-bound regions: HIF1α binding sites showed 1.37-fold higher average levels of the promoter mark H3K4me3, whereas levels of the enhancer mark H3K4me1 were 0.75-fold lower at HIF1α binding sites than at HIF2α sites (Additional file 1: Fig. S3i). Similarly, chromHMM analysis showed enrichment of HIF1α at promoters and depletion at enhancers relative to HIF2α (Additional file 1: Fig. S3j). Moreover, other TFs similarly differed in occupancy between HIF1α- and HIF2α-specific sites: HIF2α was enriched at MCF7-specific TF binding sites (which mostly correspond to cell-type-specific enhancers), and TFs shared between MCF7, RCC4, and SK-MEL-28 showed no enrichment of binding between HIF1α and HIF2α target sites (Additional file 1: Fig. S3f,k). In conclusion, the binding preferences of HIF1α and HIF2α differ, with HIF1α being somewhat more promoter-enriched and HIF2α being more enhancer-enriched, but these preferences are not determined by differences in DNA methylation at their binding sites.

### DNA methylation directly repels HIF binding in cells

To more firmly establish a causal link between DNA methylation and HIF binding, we excluded several confounders. Firstly, since our chromatin state analysis revealed that HIF preferentially binds active enhancers and promoters, which are known to carry low levels of methylation [24], we performed HIF1β ChIP-bisulfite sequencing (HIF1β ChIP-BS-seq). MCF7 cells were exposed to hypoxia; HIF1β-bound DNA was immuno-precipitated and bisulfite-converted prior to sequencing to uncover its methylation pattern. This revealed that, while methylation levels of input DNA (not immunoprecipitated, bisulfite-converted DNA) were mostly low but with some sites displaying intermediate to high methylation levels, HIF1β-bound DNA was invariably very low in methylation and this at all sites (Fig. 2a, Additional file 1: Fig. S4a).

Secondly, since TFs can drive demethylation of their binding sites both passively and actively, we excluded the possibility that DNA fragments bound by HIF would undergo DNA demethylation upon HIF binding. Indeed, HIF1β has previously been shown to actively recruit DNA demethylases [26]. However, HIF1β ChIP-BS-seq in hypoxic ESCs deficient for all DNA demethylases (*Tet1*, *Tet2*, and *Tet3*) showed results identical to those observed in wild-type MCF7 cells: HIF1β-bound DNA was unmethylated compared to input DNA subjected to whole-genome BS-seq (Fig. 2b, Additional file 1: Fig. S4b).

Additionally, other (unknown) confounders related to the binding location of HIF, such as chromatin environment or sequence context, may contribute to preferential HIF binding to unmethylated DNA. To exclude this possibility, we generated isogenic murine ES cell lines in which a human HIF1β binding site-encoding DNA fragment was inserted that was either in vitro methylated or not (Fig. 2c). Following recombination, the difference in methylation state between both fragments was maintained (Additional file 1: Fig. S4c-d). HIF1β ChIP-qPCR revealed that methylation was sufficient to induce a 12.4-fold reduction in HIF1β binding in these isogenic cell lines (Fig. 2d).

Finally, to directly assess methylation sensitivity of HIF binding to unchromatinized DNA, we employed microscale thermophoresis and tested the binding of recombinant co-purified HIF1α-HIF1β and HIF2α-HIF1β heterodimers to double-stranded DNA oligonucleotides containing a methylated or unmethylated RCGTG motif. Importantly, HIF1α- and HIF2α-containing heterodimers both showed a 15-fold higher affinity ($K_D$)

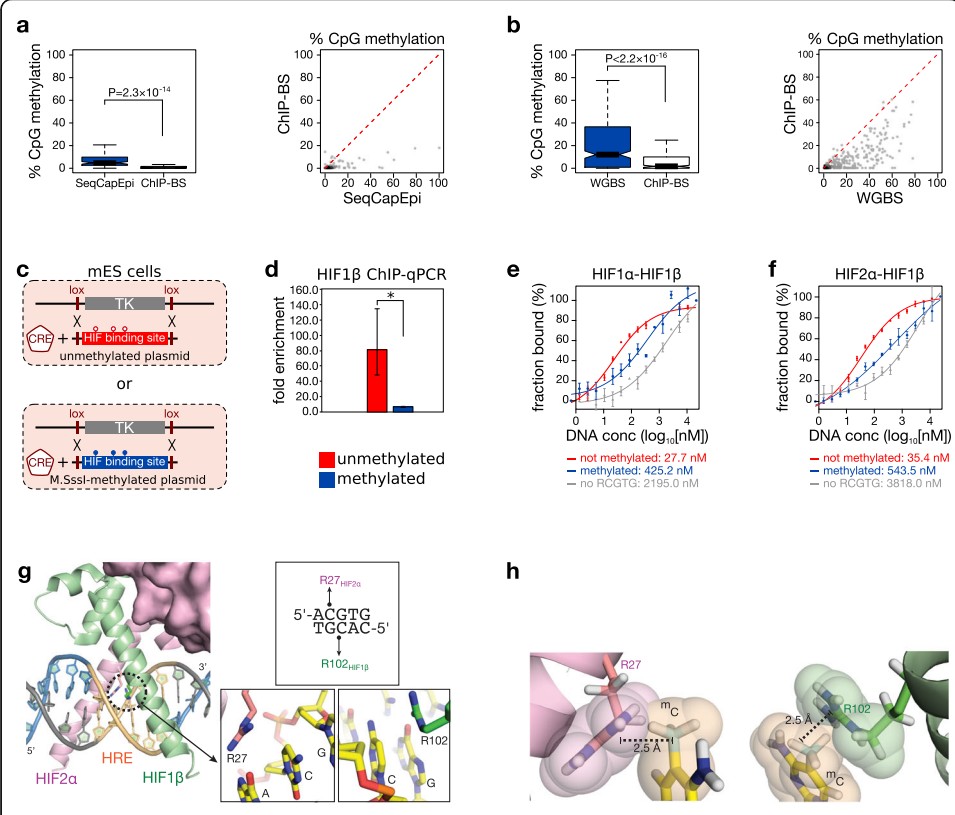

**Fig. 2** DNA methylation directly repels HIF1β binding. **a, b** Boxplot (*left*) and scatter plot (*right*) of methylation levels of HIF1β-bound immunoprecipitated DNA fragments obtained by ChIP-BS-seq (ChIP-BS) compared to input by SeqCapEpi BS-seq (SeqCapEpi) in MCF7 cells (**a**), or of HIF1β-bound immunoprecipitated DNA fragments obtained by ChIP-BS compared to input by whole-genome BS-seq (WGBS) in mouse *Tet*-triple-knockout (*Tet*-TKO) ESCs (**b**). The red dotted line in the scatter plot indicates the theoretical value of equal methylation in immunoprecipitated and input DNA. *P* values by *t*-test. **c, d** Recombination-mediated cassette exchange. **c** A human HIF binding site (chr16: 30,065,212-30,065,711 on hg38) was cloned between 2 L1 *Lox* sites and in vitro methylated (*blue*) or not (*red*) prior to co-transfection with a CRE recombinase-encoding plasmid into mESCs transformed to contain an L1 *Lox*-flanked thymidine kinase (TK). **d** Following successful cassette exchange, these ESCs were incubated in hypoxia (0.5% $O_2$ for 16 h) and probed using HIF1β ChIP-qPCR for HIF binding at the differentially methylated cassette. Shown is the fold enrichment over background ($n = 3$ independent ChIP pairs; *$P < 0.05$ by *t*-test). **e, f** Microscale thermophoresis-based assessment of sensitivity of HIF1α-HIF1β (**e**) and HIF2α-HIF1β (**f**) heteroduplexes to methylation at HIF binding sites in physiological buffer (PBS). RCGTG sequences in the double-stranded DNA oligonucleotides were either absent (*gray*), methylated (*blue*), or unmethylated (*red*) at the CpG site. Calculated $K_D$ values are shown under each graph. **g** Excerpt from the crystal structure of HIF2α-HIF1β in complex with a DNA duplex containing the core HIF binding sequence 5'-ACGTG-3' (PDB code 4ZPK) [25]. **h** Modeling of methylation of CpG cytosines in ACGTG reveals severe steric hindrance. The two views show hard-sphere models of methylated cytosines modeled at position 5 (including bonding hydrogen atoms) and how they severely violate the van der Waals envelopes (2.5 Å width) of Arg27 in HIF2α (*left*) and Arg102 in HIF1β (*right*)

for an unmethylated than methylated RCGTG motif, thus confirming that methylation directly repels binding of HIF1α-HIF1β and HIF2α-HIF1β heterodimers (Fig. 2e, f). Indeed, leveraging the crystal structure of the HIF1α-HIF1β and the HIF2α-HIF1β complexes bound to DNA [25], revealed that both cytosines in the CpG dinucleotide of the HIF binding sequence are snuggly accommodated via van der Waals interactions with the guanidine groups of Arg102 in HIF1β and Arg27 in HIF1α or HIF2α, respectively (Fig. 2g) [25]. Methylation of any of the two cytosines either on the top or bottom strand would in a static model drastically violate the minimum 3.1 Å length of van der

Waals radii and would be poised to cause severe steric clashes with these two functionally important arginine residues in HIF1α or HIF2α (Fig. 2h).

### DNA demethylation enables ectopic HIF binding

Next, we investigated which parts of the genome are protected from HIF binding by DNA methylation. For this, we compared HIF1β binding in hypoxic wild-type murine ESCs versus ESCs deficient for DNA methyltransferases (*Dnmt*-TKOs), which lack DNA methylation [27], using HIF1β ChIP-seq ($n = 4$ replicates for each; for data quality assessment see Additional file 1: Fig. S5). This revealed a marked increase in the number of HIF1β binding peaks, from 7875 in wild-type to 9806 in *Dnmt*-TKO ESCs (Fig. 3a). Whole-genome BS-seq further revealed that, while shared binding peaks were unmethylated in both cell lines, *Dnmt*-TKO-specific HIF1β binding peaks had high methylation levels in wild-type ESCs (Fig. 3b).

All shared binding peaks were associated with a similar enrichment of the RCGTG motif (Fig. 3c), as well as with genes that were induced upon hypoxia (Fig. 3d). However, *Dnmt*-TKO-specific sites were more often distal to annotated transcription start sites (TSS) or regions of open chromatin, and more frequently in repressed chromatin regions of wild-type ESCs (Fig. 3e–g). Gene ontology analysis moreover failed to identify enrichment of hypoxia-related processes for *Dnmt*-TKO-specific binding peaks, in contrast to shared peaks (Fig. 3h). Thus, the majority of these *Dnmt*-TKO-specific binding peaks represents ectopic binding events.

### DNA methylation represses hypoxia-induced expression of retrotransposons

Indeed, a substantial fraction of novel *Dnmt*-TKO-specific HIF1β binding peaks were found in repetitive genomic regions. Particularly, repeat class analysis revealed a 1.65-fold increase in binding peaks near retrotransposons (2737 of 7875 (34.8%) shared peaks versus 1106 of 1931 (57.3%) *Dnmt*-TKO-specific peaks; Additional file 1: Fig. S6a). Although HIF1β binding events were frequently observed at LINEs and SINEs, only binding at long terminal repeats (LTRs) was enriched over a randomization of HIF1β binding site positions, and this both for all binding events and those distal to TSS (Fig. 3i). The bulk of this increase was ascribable to binding at the 5′-end of endogenous retrovirus K (ERVK) LTR sequences (Fig. 3j), with 344 of 1106 (31%) novel repeat-binding peaks being at ERVKs versus only 3% of randomly shuffled HIF1β binding sites. These were mostly at solitary LTRs (Additional file 1: Fig. S6b-e). Given that ChIP-seq analyses rely on uniquely mapping reads, which are inherently depleted at repeat regions, this enrichment is likely to represent an underestimate.

We then assessed whether a similar phenomenon is at play in cancer cell lines, and pharmacologically demethylated MCF7 cells using a non-cytotoxic [28] dose of 5-aza-2′-deoxycytidine (aza, 1 μM), necessary and sufficient for strongly reducing DNA methylation (Additional file 1: Fig. S6f). HIF1β ChIP-seq revealed that aza exposed 1236 new HIF1β binding peaks. These were all methylated in untreated MCF7 cells and showed a 2.5-fold reduced methylation in aza-treated cells (Fig. 4a-b). While HIF1β binding peaks in retrotransposons were already present in vehicle-treated MCF7 cells, novel aza-specific HIF1β binding peaks were 1.7-fold enriched for retrotransposons (9.7% versus 16.4%, respectively; Additional file 1: Fig. S6g). Again, these novel HIF1β

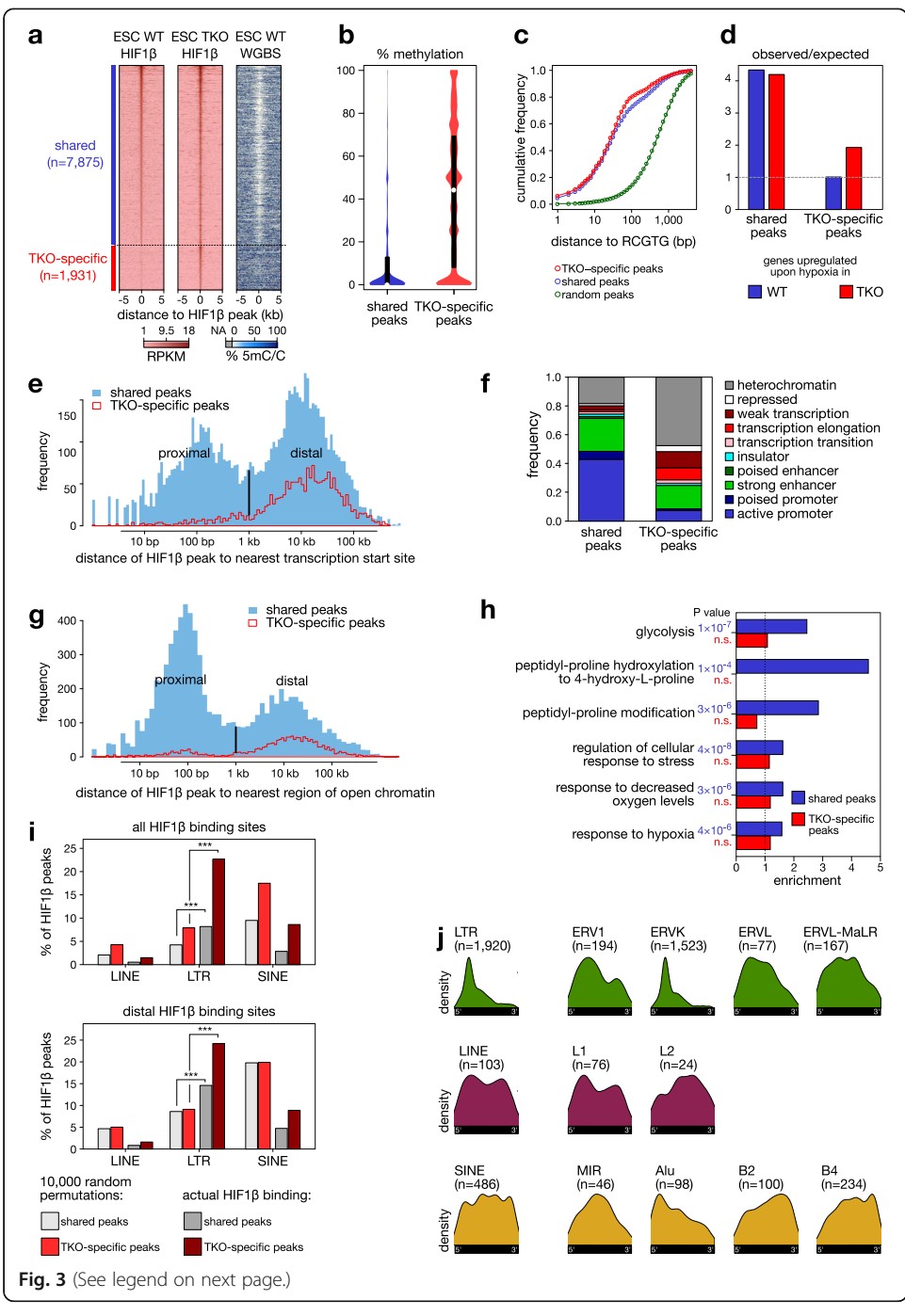

**Fig. 3** (See legend on next page.)

(See figure on previous page.)

**Fig. 3** DNA demethylation uncovers new HIF1β binding sites. **a** Heatmaps of HIF1β binding (RPKM) and DNA methylation as determined using WGBS at regions flanking the summit of HIF1β binding peaks (± 5 kb) either shared with WT or TKO-specific ESCs. **b** % methylation at shared and TKO-specific HIF1β binding sites in WT ESCs. See Additional file 1: Fig. S5 for scatter plots illustrating the correlations between HIF1β ChIP-seq replicates in *Dnmt*-WT versus *Dnmt*-TKO ESCs. **c** Cumulative frequency of distance to the nearest RCGTG motif for shared, TKO-specific, and randomized HIF1β binding peaks. **d** Observed/expected frequency of upregulated genes associated with shared and TKO-specific HIF1β binding peaks in WT and *Dnmt*-TKO ESCs exposed to 24 h of hypoxia (0.5% $O_2$). **e** Distance of shared and TKO-specific HIF1β binding peaks in ESCs to the nearest TSS. A bimodal peak was detected indicating proximal and distal binding events. **f** Functional genome annotation using ChromHMM of shared and TKO-specific HIF1β binding peaks in ESCs. **g** Distance of shared and TKO-specific HIF1β binding peaks to open chromatin regions in ESCs. A bimodal peak was detected indicating proximal and distal binding events. **h** Ontology analysis of genes associated with shared and TKO-specific HIF1β binding peaks in ESCs. **i** HIF1β binding sites in LINEs, LTRs, and SINEs after 10,000 random permutations and as observed by HIF1β ChIP-seq (actual HIF1β binding) for all HIF1β sites (*top panel*) and only for distal HIF1β sites (*bottom panel*). *** $P < 0.001$ by Fisher's exact test. **j** Distribution of HIF1β binding peaks detected in murine *Dnmt*-TKO ESCs for the retrotransposon families, color-coded by retrotransposon class (green: LTR; violet: LINE; yellow: SINE)

binding peaks were often distal to TSSs, and binding at LTRs was enriched over a randomization of HIF1β binding site positions (Fig. 4c). Notably, different retrotransposons were affected in human MCF7 cells compared to murine ESCs due to the evolutionarily divergent repeat content of these genomes. An analysis of the distribution of HIF1β binding peaks at retrotransposons, however, revealed that HIF1β binding sites were often at the 5′-end of retrotransposon sequences and that patterns of binding were conserved between mouse and human genomes (Fig. 3j versus Fig. 4d), suggesting that HIF binding on retrotransposons is not random but functional, inducing their expression.

To confirm the latter, we applied RNA-seq to assess changes in retrotransposon expression after 24 h of hypoxia with or without aza. Repeat expression was analyzed using different bioinformatics pipelines. First, we used RepEnrich [29], which combines repeat-associated reads, also those that are non-uniquely mapping, to assess repeat expression for each of the 779 retrotransposon subfamilies annotated in the human genome (these are each member of one of the 25 families that constitute the LTR, LINE, and SINE retrotransposon classes). We found that, already under hypoxia alone, 251 of all LTR (44%), 51 of all LINE (32%), and 5 of all SINE (10%) subfamilies were upregulated, while only 16 LTR, 7 LINE, and no SINE subfamilies were downregulated (5% FDR; Additional file 1: Fig. S6h-i). Next, we used SQuIRE, which assigns reads (including non-uniquely mapping reads) to a specific repeat locus based on an expectation-maximization algorithm [30]. With SQuIRE, 72% ($n = 2781$) of all differentially expressed repeat loci exhibited increased expression under hypoxia ($P < 10^{-16}$; Additional file 1: Fig. S6j).

### Induction of cryptic transcripts by hypoxia

Hypoxia-induced transcripts were, however, often not matching the annotated repeat locus, but extending well beyond their annotated end, with some transcripts encompassing multiple repeat elements. Also, we noticed that for many transcripts, HIF1β binding did not occur in the retrotransposon promoter, while some other transcripts did even not contain a retrotransposon-associated sequence. Similar transcripts were also induced by aza. We therefore refer to these as "cryptic transcripts" (Additional file 1: Fig. S7a). To more accurately quantify them, we developed a novel analysis pipeline,

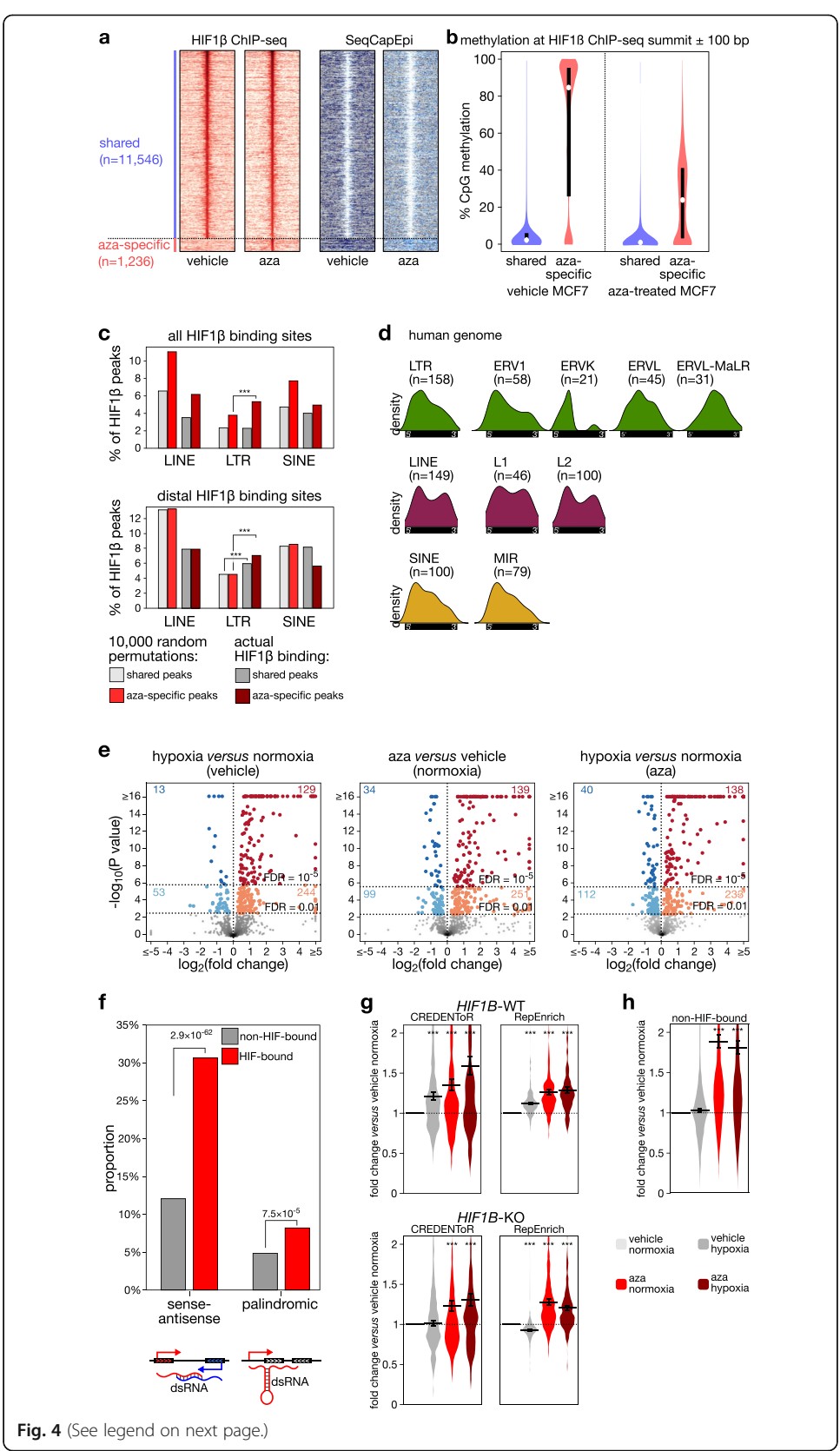

**Fig. 4** (See legend on next page.)

(See figure on previous page.)

**Fig. 4** DNA methylation represses hypoxia-induced cryptic transcript activation (**a**) Heatmaps of HIF1β binding (RPKM) and DNA methylation as determined using SeqCapEpi BS-seq at regions flanking the summit of HIF1β binding peaks (± 5 kb). Shown are HIF1β binding peaks that are shared between vehicle-treated and aza-treated MCF7 cells, or that are specific to aza-treated cells. In total, 12,782 HIF1β binding peak positions were detected across vehicle- and aza-treated MCF7 using a $P < 10^{-15}$ threshold. **b** Violin plots of methylation detected by SeqCapEpi BS-seq at HIF1β binding peaks that are shared between vehicle-treated and aza-treated MCF7 cells, or that are specific to aza-treated MCF7 cells. **c** HIF1β binding sites in LINEs, LTRs, and SINEs after 10,000 random permutations and as observed by HIF1β ChIP-seq (actual HIF1β binding) for HIF1β binding peaks that are shared between vehicle- and aza-treated MCF7 cells, or specific to aza-treated MCF7 cells for all HIF1β sites (*top panel*) and only for distal HIF1β sites (*bottom panel*). **d** Distribution of HIF1β binding peaks detected in aza-treated MCF7 cells at retrotransposon families, color-coded by retrotransposon class (green: LTR; violet: LINE; yellow: SINE). **e** Volcano plots showing differential expression of HIF-bound cryptic transcripts, as determined by CREDENToR in MCF7 cells exposed to vehicle (DMSO) or 5-aza-2′-deoxycytidine (aza; 1 μM) for 4 days, hypoxia (0.5% oxygen, 1 day) or normoxia. Significantly upregulated and downregulated transcripts are highlighted in red and blue, respectively. The associated numbers refer to how many transcripts are up- or downregulated at a 1% FDR and a 0.001% FDR, as indicated by the horizontal line. **f** dsRNA formation potential of all cryptic transcripts (*gray*) and of HIF-bound cryptic transcripts (*red*). Shown are the fraction of all RNAs for which transcription overlaps with a transcript expressed from the complementary strand ("sense-antisense"), and RNAs containing the same retrotransposon repeat element in sense and antisense orientation ("palindromic"). *P* values from the chi-square test. **g**, **h** Expression of HIF-bound (**g**) and non-HIF-bound (**h**) cryptic transcripts relative to vehicle-treated controls (vehicle normoxia) in MCF7 cells wild-type (WT) (**g** *upper panels* and **h**) or knockout (KO) (**g** *bottom panels*) for *HIF1B*. Shown are expression changes as assessed using CREDENToR (**g** *left panels* and **h**) and RepEnrich (**g** *right panels*), with error bars indicating geometric mean ± s.e.m. n.s. not significant, ***P < 0.001 by *t*-test

CRyptic Elements' Differential Expression by de Novo Transcriptome Reconstruction (CREDENToR). CREDENToR first performs a de novo transcriptome assembly to define cryptic transcripts and then assigns uniquely mapping reads to them to quantify their expression. The cryptic transcripts detected by CREDENToR are poorly conserved, often unspliced transcripts, shorter than lincRNAs but expressed at similar levels (see "Methods" and Additional file 1: Fig. S7b-g for benchmarking).

CREDENToR identified that out of 1389 differentially expressed cryptic transcripts (1% FDR), 67% were upregulated by hypoxia (Additional file 1: Fig. S6k). As expected, focusing on HIF-bound cryptic transcripts revealed an even stronger enrichment, with 82% and 91% (respectively, at 1% and 0.001% FDR) differentially expressed transcripts being upregulated following hypoxia (Fig. 4e). HIF binding was enriched at the promoter of hypoxia-induced cryptic transcripts, but far less in those induced by aza (Additional file 1: Fig. S6l). Interestingly, significant fractions of cryptic transcripts contained palindromic repeats, or overlapped with other transcripts in the reverse orientation, and could thus produce double-stranded (ds) RNA. HIF-bound cryptic transcripts were twice as likely to generate such dsRNAs (Fig. 4f). Together, this suggests HIF binding to leverage cryptic TSS structures within and outside the repeat genome to express dsRNA-generating cryptic transcripts.

Cryptic transcript expression was indeed dependent on HIF, as non-HIF-bound cryptic transcripts failed to show induction following hypoxia (Fig. 4g). To confirm this, we assessed expression in *HIF1B*-knockout MCF7 cells. Here, hypoxia failed to upregulate cryptic transcripts, according to both CREDENToR and RepEnrich (Fig. 4g, Additional file 1: Fig. S6m). As expected, aza-induced overexpression was retained, while hypoxia in *HIF1B*-knockout MCF7 cells failed to increase the effect of aza. Pharmacological activation of HIF using dimethyloxalylglycine (DMOG), a broad-spectrum inhibitor of 2-oxoglutarate-dependent hydroxylases [31], affected cryptic transcripts similar to

hypoxia (Additional file 1: Fig. S6m-o). Combined, these data indicate that hypoxia triggers HIF binding to unmethylated repeat regions, inducing HIF-dependent expression of cryptic transcripts, most of which are associated with retrotransposons.

### Hypoxia and repeat transcript expression affect tumor immunotolerance

Expression of repeat transcripts has been linked to tumor foreignness [32], interferon (IFN) response [33–35], and enhanced cytolytic activity [36], all critical determinants of response to checkpoint immunotherapy. Similar to our own data, such transcripts were shown to increase dsRNA formation. This triggers IFN responses through viral mimicry. Cryptic transcripts induced by HIF could thus contribute to an immune-activated microenvironment.

To study this in more detail, we reanalyzed expression and DNA methylation data from The Cancer Genome Atlas (TCGA). We classified 5193 tumors from 14 tumor types as hypoxic or normoxic using an established hypoxia metagene expression signature [37]. We remapped all RNA-seq reads to determine expression of retrotransposon subfamilies using RepEnrich, and also performed de novo transcript assembly to identify on average 11,654 non-overlapping cryptic transcripts per tumor type using CREDENToR (Additional file 1: Fig. S8a). While TCGA tumors were not exposed to DNA demethylating agents, they did show variation in DNA methylation at TSS of cryptic transcripts. Indeed, although CpGs in cryptic transcript promoters showed mostly high methylation levels (median = 80.7%), there was considerable variability (9.2% standard deviation), and one in 10 tumors displayed median levels below 67.3%. Remarkably, and in line with our in vitro data, there was a significant interaction between hypoxia and DNA methylation in determining cryptic transcript expression ($P = 0.0109$), with expression being increased in hypoxic tumors having lower methylation at cryptic transcripts (Fig. 5a). At least 1279 cryptic transcripts showed increased expression of 10-fold or higher (FDR < 0.01, Additional file 1: Fig. S8b). A reanalysis of combined single-cell methylome-and-transcriptome sequencing data from colorectal cancer cells [39] moreover confirmed that cryptic transcript expression and promoter methylation are inversely correlated, and this more strongly in hypoxic than normoxic cancer cells ($P = 0.032$ in a general linear model), suggesting that the observed interactions are cancer cell-intrinsic (Additional file 4).

In TCGA, this interaction was only detected in tumor types known to respond to immunotherapy [38] ($P = 0.0031$ in $n = 2505$ responsive tumors versus $P = 0.69$ in $n = 2681$ non-responsive tumors; see "Methods" for a detailed description of the generalized linear model; Fig. 5b, c). As expected, responsive tumor types exhibited an increased tumor mutation burden (TMB), elevated immune checkpoint expression, more CD8$^+$ T cells, and increased cytolytic activity (Additional file 1: Fig. S8c) [40]. Importantly, responsive types also had on average lower methylation at cryptic transcripts and higher cryptic transcript expression than non-responsive types ($P < 10^{-16}$ for both comparisons, Fig. 5d). Single-cell RNA-seq analyses (both from 5′ and 3′ end) highlighted that cancer cells show the highest level of cryptic transcript expression compared to stromal cells, indicating they represent the main source of cryptic transcripts expression (Additional file 1: Fig. S8d). In line with our in vitro findings, DNA hypomethylation thus underlies cryptic transcript expression in hypoxic tumors, an effect that was particularly striking in immunotherapy-responsive tumors.

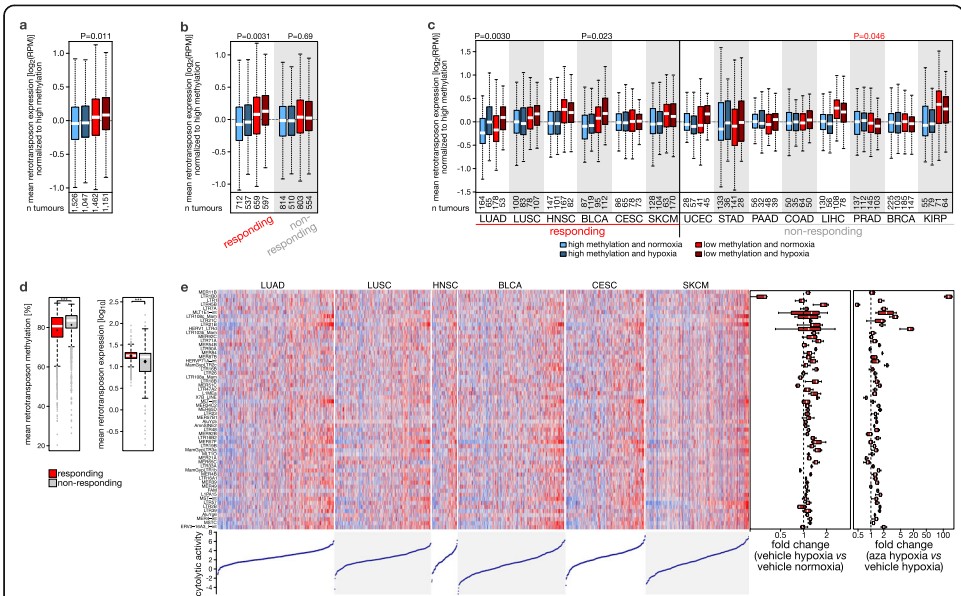

**Fig. 5** Cryptic transcript expression in tumors. **a–c** Cryptic transcript expression in tumors characterized by TCGA. Shown is cryptic transcript expression in tumors with high or low methylation of cryptic transcript promoter regions (blue or red; > or ≤ the median methylation level of each tumor type), and in normoxic or hypoxic (light or dark color) tumors. Data are shown for **a** all tumor types combined, **b** stratified into those that are responding or non-responding to immunotherapy following the classification described by Turajlik and colleagues [38], and **c** for each tumor type separately. *P* values by *t*-test, red values indicating inverse correlations. **d** DNA methylation levels at cryptic transcript promoters (*left*) and cryptic transcript expression (*right*) in tumors profiled in TCGA, stratified into tumor types that are responsive (*n* = 2280) or non-responsive (*n* = 2214) to checkpoint immunotherapy. ****P* < 0.001 by *t*-test. **e** Heatmap showing the expression (*Z*-score, blue to red) of the 59 cryptic transcripts associated with cytolytic activity in tumors responsive to immunotherapy from TCGA. The boxplot on the right depicts the log fold change in expression of the same 59 cryptic transcripts in hypoxic versus normoxic MCF7 cells (24 h, 0.5% O₂), and of MCF7 cells after 4-day exposure to aza versus vehicle-treated hypoxic MCF7 cells (*P* < 0.05 for all cryptic transcripts, either for hypoxia versus vehicle, or for hypoxia plus aza versus aza alone). At the bottom, cytolytic activity of each TCGA sample is depicted. LUAD; lung adenocarcinoma; LUSC, lung squamous cell carcinoma; HNSC, head and neck squamous cell carcinoma; BLCA, bladder urothelial carcinoma; CESC, cervical squamous cell carcinoma and endocervical adenocarcinoma; SKCM, skin cutaneous melanoma

Overall, these observations support a model wherein hypoxia-induced cryptic transcripts are tolerated in high-immunogenic tumors, as these are characterized by high immune checkpoint expression, but not in low-immunogenic tumors where their expression would compromise tumor immunotolerance. This suggests that low-immunogenic tumors may need to maintain high DNA methylation levels in cryptic transcripts to downregulate their expression and avoid the induction of tumor immunogenicity.

## Aza compromises tumor immunotolerance in mice via HIF

To confirm that in low-immunogenic tumors DNA methylation prohibits cryptic transcript expression, we identified 59 such retrotransposons that correlate in expression with cytolytic activity in immunotherapy-responsive tumor types within TCGA. Remarkably, all of these were upregulated in vitro, by hypoxia alone or hypoxia in combination with aza (*P* < 0.05, Fig. 5e), suggesting that hypoxia and DNA demethylation can indeed enhance tumor immunogenicity. To confirm this experimentally, we screened several mouse tumor models for their immunogenicity. The orthotopic 4T1

breast cancer model was identified as low-immunogenic. Indeed, 4T1 tumors exhibited a low TMB, cytolytic activity, number of $CD8^+$ T cells and expression of immune checkpoints (*Pd1*, *Pdl1*) compared to other models (Additional file 1: Fig. S9a). In line with 4T1 grafts being low-immunogenic tumors, anti-PD1 treatment failed to affect their growth (– 8%, $P = 0.397$), while significantly reducing growth of high-immunogenic tumors, as described previously [41, 42] (Additional file 1: Fig. S9b). Importantly, also the expression of cryptic transcripts was lower in 4T1 than in high-immunogenic tumor models (Additional file 1: Fig. S9c).

Next, we verified in low-immunogenic 4T1 cells whether DNA demethylation upregulates cryptic transcripts in a HIF-dependent manner. In vitro, we observed that, similar to MCF7 cells, both hypoxia and aza independently increased cryptic transcript expression, both using CREDENToR and RepEnrich (Fig. 6a; Additional file 1: Fig. S9d). Likewise, aza increased cryptic transcript expression in vivo (Fig. 6b). To confirm that this upregulation was at least partially hypoxia-mediated, we investigated whether tumor hypoxia enhances aza-induced cryptic transcript expression. We compared aza-treated 4T1 tumor-bearing mice injected either with control or anti-VEGFR-2 antibody (DC101). While vehicle-treated 4T1 tumors were hypoxic in ~ 40% of the tumor, DC101 further reduced blood vessel density (– 35%; $P < 0.05$) and increased hypoxic tumor areas (68%; $P < 0.05$; Fig. 6c, d). Importantly, this was associated with an increase in cryptic transcripts (+ 9%; $P = 2.6 \times 10^{-16}$; Fig. 6e).

We then explored whether this increase also compromised immunotolerance. As immunogenicity of cryptic transcripts is mediated via dsRNA formation, we first confirmed in vitro by immunostaining the increase in dsRNA after both hypoxia and aza in 4T1 cells (Fig. 6f). In vivo, aza reduced growth of 4T1 tumors (– 32%; $P = 3.0 \times 10^{-3}$; Fig. 6g), but did not reduce cell proliferation marker expression (Additional file 1: Fig. S9e). In contrast, immune activation was enhanced in tumors treated with aza, as activated T cell and natural killer cell signatures were upregulated and myeloid-derived suppressor cell signatures downregulated (Fig. 6h). Immunofluorescence of $CD8^+$ T cells confirmed these changes: while T cell infiltration was unaffected, the number of activated, granzyme B-positive T cells increased 2.1-fold ($P < 0.05$; Additional file 1: Fig. S9f).

To verify HIF-dependence of these immunogenic effects, we generated polyclonal 4T1 cells deficient for HIF1β by CRISPR-Cas9 ($4T1^{Hif1b\text{-}KO}$; Additional file 1: Fig. S9g) and compared these cells to scramble-control 4T1 cells ($4T1^{Hif1b\text{-}scr}$), while treating with aza or vehicle. In vitro, loss of HIF1β abrogated hypoxia-induced dsRNA formation and HIF1β-bound cryptic transcript expression both in vehicle and aza-treated cells (Additional file 1: Fig. S9d and h), similar to what we observed in MCF7 cells. Also in vivo, $4T1^{Hif1b\text{-}KO}$ showed reduced cryptic transcript expression compared to $4T1^{Hif1b\text{-}scr}$ grafts, effects that were limited to HIF-bound cryptic transcripts as expected (Fig. 6i; Additional file 1: Fig. S9i). Of note, $4T1^{Hif1b\text{-}KO}$ tumors grew much more slowly than $4T1^{Hif1b\text{-}scr}$ tumors, presumably because HIF1β also has direct effects on cell proliferation, thus rendering it challenging to disentangle effects on immunogenicity. Nevertheless, aza induced a similar and significant upregulation of cancer testis antigen expression in both cell lines (Additional file 1: Fig. S2), suggesting similar treatment efficacy. Interestingly, while $4T1^{Hif1b\text{-}scr}$ grafts also showed a significantly reduced size when comparing aza to vehicle (46% reduction; $P = 3.3 \times 10^{-6}$), $4T1^{Hif1b\text{-}KO}$ failed to show as strong a reduction (only 22%; $P = 7.0 \times 10^{-4}$, or 1.8-fold less than the scramble

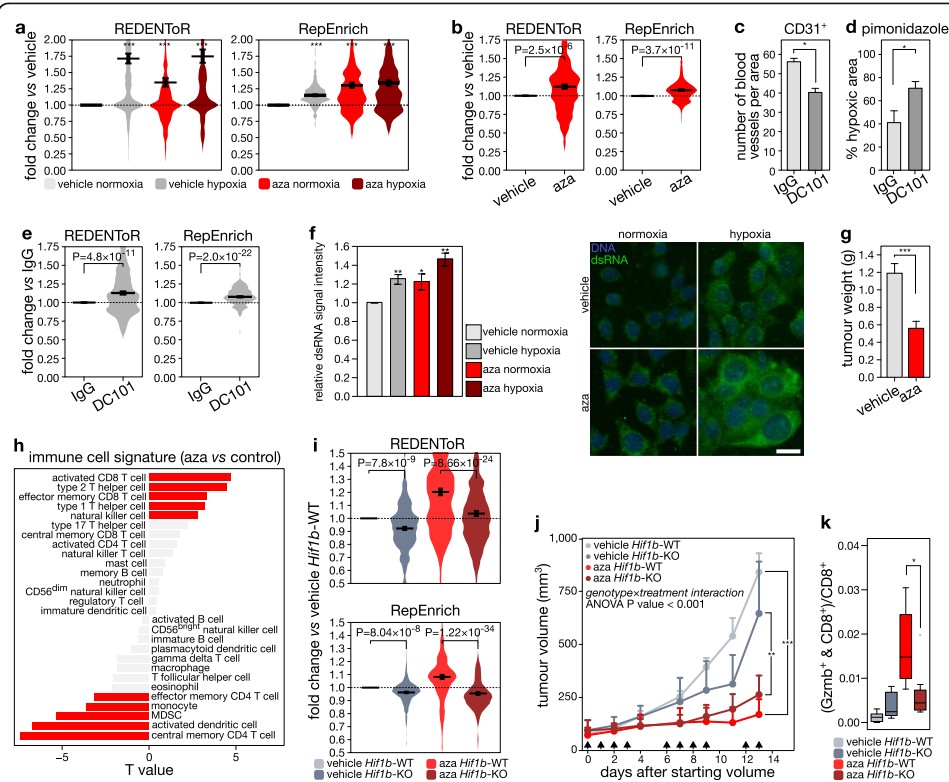

**Fig. 6** Aza treatment increases tumor immunogenicity HIF-dependently. **a** Expression in 4T1 cells of cryptic transcripts (CREDENToR, *left*) or retrotransposon subfamilies (RepEnrich, *right*) bound by HIF1β in hypoxic 4T1 cells, following exposure to vehicle (DMSO) or 5-aza-2′-deoxycytidine (aza; 1 μM) for 4 days, hypoxia (0.5% oxygen, 1 day) or normoxia. Difference in the distribution of expression is expressed as fold change of counts per million over control 4T1 cells. Error bars indicate geometric mean ± s.e.m. **b** Expression of cryptic transcripts (CREDENToR, *left*) or retrotransposon subfamilies (RepEnrich, *right*) bound by HIF1β in 4T1 cells in vehicle- and aza-treated 4T1 tumors (*n* = 6 per treatment condition). Difference in the distribution of expression is expressed as fold change of counts per million over control 4T1 tumors. Error bars indicate geometric mean ± s.e.m. **c**, **d** Quantification of the number of blood vessels (CD31 staining; **c**) and percentage of hypoxia (pimonidazole staining; **d**) in 4T1 tumors from mice injected with DC101 or control IgG (with at least 4 mice per treatment condition, see "Methods"). **e** Expression of cryptic transcripts and retrotransposon subfamilies (bound by HIF1β in 4T1 cells) as determined by CREDENToR (*left*) and RepEnrich (*right*) in control antibody- and DC101-treated 4T1 tumors (*n* = 6 per treatment condition). Difference in the distribution of expression is expressed as fold change of counts per million over control 4T1 tumors. Error bars indicate geometric mean ± s.e.m. **f** (*left*) Signal intensity of dsRNA staining in 4T1 cells treated with aza or PBS and incubated in hypoxia or normoxia for 24 h. (*right*) Immunofluorescence of dsRNA using a dsRNA antibody (clone J2, green) in 4T1 cells treated with aza or vehicle (PBS), and by a 24-h incubation in hypoxic (0.5% O₂) or normoxic conditions (scale 40 μm). A representative image is shown for each condition. **g** Barplot showing the tumor weight of vehicle- and aza-treated 4T1 tumors (*n* = 6 per treatment condition). ****P* < 0.001 by paired *t*-test. **h** Association between aza treatment and immune cell infiltration estimates in 4T1 tumors from mice treated with either aza or PBS, as calculated by GSVA on PanCancer immune metagenes [43] and visualized by their *T* value (at least 6 mice per treatment condition were sequenced). Red bars indicate significant associations (*P* < 0.05). **i** Expression of cryptic transcripts (CREDENToR, *top*) and retrotransposon subfamilies (RepEnrich, *bottom*) bound by HIF1β in 4T1 cells in 4T1 tumors WT or KO for *Hif1b* implanted in mice treated with vehicle or aza (see "Methods," at least 6 tumors per treatment condition were sequenced). Difference in the distribution of expression is expressed as fold change of counts per million over *Hif1b*-WT vehicle-treated 4T1 tumors. Error bars indicate geometric mean ± s.e.m. **j** Growth of tumors generated by grafting mice orthotopically with 4T1 cells wild-type (4T1^{Hif1b-scr}) or KO for *Hif1b* (4T1^{Hif1b-KO}). Mice were treated with aza or vehicle (PBS) on the days indicated with an arrow (see "Methods"). Data represent mean and s.e.m. from independent experiments each with at least *n* = 6 mice per group. **P* < 0.05 by *t*-test. A genotype-by-treatment interaction as assessed by ANOVA was *P* < 0.001. **k** Quantification of CD8⁺ and granzyme b (Gzmb)⁺ cells, depicted as percentage of CD8⁺ cells, from 4T1 cells WT for *Hif1b* (4T1^{Hif1b-scr}) or KO for *Hif1b* (4T1^{Hif1b-KO}) and treated with aza or vehicle (PBS) (*n* = 6 per group; see "Methods"). **P* < 0.05 by *t*-test

effect; $P = 0.021$; Fig. 6j). The differential effect of aza in 4T1$^{Hif1b\text{-KO}}$ versus 4T1$^{Hif1b\text{-scr}}$ grafts was also highly significant in an interaction analysis ($P < 0.0001$). Moreover, while the number of activated T cells increased in 4T1$^{Hif1b\text{-scr}}$ grafts following aza, 4T1$^{Hif1b\text{-KO}}$ grafts failed to show such increase (Fig. 6k). This differential effect of aza, depending on the *Hif1b* background, was similarly significant in an interaction analysis ($P = 6.3 \times 10^{-3}$). Together, these data provide a mechanistic link between HIF1β binding, DNA methylation, and immune activation, highlighting the potential of DNA methylation inhibitors to activate the immune system and render immune-cold tumors immune-hot.

## Discussion

Here, we show that DNA methylation directly repels binding of HIF transcription factors and that cell-type-specific DNA methylation patterns established under normoxic conditions underlie the differential hypoxic response between cell types. Furthermore, ectopic HIF binding sites in repeat elements are normally masked by DNA methylation but become accessible to HIF upon DNA demethylation, leading to expression of cryptic transcripts which enhance tumor immunogenicity.

Our findings are important for a number of reasons. Firstly, an instructive role of DNA methylation in gene expression regulation, as originally proposed by Holliday and Pugh and by Riggs [44, 45], has remained controversial. Indeed, in many instances it is unclear whether DNA methylation changes are a direct or indirect cause, or rather a consequence of TF binding or gene expression [2]. Our findings in murine and human both differentiated and undifferentiated cells align well with a recent study showing methylation dependence of NRF1 binding in mESCs. By demonstrating that DNA methylation directly repels HIF binding, we thus highlight the importance of DNA methylation profiling, especially in poorly oxygenated tissues. Since tumor hypoxia has long been associated with increased malignancy, poor prognosis, and resistance to radio- and chemotherapy [6], DNA methylation could especially provide insights in the processes underlying therapeutic resistance. For instance, Vanharanta and colleagues recently showed an association between DNA methylation near *CYTIP* and the survival of disseminating cancer cells [46]. Combined with our observations that DNA methylation directly repels HIF binding, this suggests remethylation of the *CYTIP* promoter as a viable avenue for decreasing cancer dissemination.

Secondly, it has been challenging to identify a guiding principle as to why specific genes are induced upon hypoxia in one, but not the other cell type [10]. Our findings suggest that cell-type-specific TF binding under normoxia causes differences in DNA methylation, which subsequently determine HIF binding under hypoxia and predict the cell-type-specific hypoxia response. We note that we did not model chronic but only acute hypoxia in vitro, conditions that do not directly alter DNA methylation and that are thus distinct from the prolonged, chronic hypoxia we previously described to be essential to cause DNA hypermethylation at promoters and enhancers by TET inhibition [1]. Importantly, we also confirmed earlier observations that HIF1β binding peaks are characterized by an active, open chromatin structure [12]. This additional requirement for functional HIF1β binding peaks probably explains why each of the RCGTG consensus sequences in the genome cannot serve as an equal HIF binding substrate in normal cells, or upon genetic or pharmacological demethylation. Similar observations were made for other TFs, such as CTCF, for which binding was similarly limited to sites

containing a permissive chromatin structure [15, 24]. Importantly, binding specificities for HIF1α versus HIF2α are independent of DNA methylation, but appear to be influenced by chromatin context. This is in line with the identical structure of DNA binding domains of HIF1α and HIF2α; swapping DNA binding domains between both proteins has no influence on their binding profile [47]. Instead, the transactivation domain appears to endow specificity, suggesting that accessory chromatin binding partners govern the differential binding of HIF1α and HIF2α [47].

Thirdly, several publications by now reported how 5-aza-2′-deoxycytidine initiates cryptic TSSs in the repeat genome, leading to expression of cryptic transcripts [33, 48]. Our data add to these findings by demonstrating that cryptic transcript expression is at least partly HIF-dependent, while more importantly, hypoxia alone is also capable of inducing their expression. Based on single-cell analyses, we observed this effect to be cancer cell-autonomous, consistent with cancer cells being hypomethylated. Our findings reinforce a growing body of evidence that highlights how during evolution transposable elements have copied and amplified regulatory regions throughout the genome [17, 49–53]. Most likely, transposable elements hijacked the transcriptional apparatus of their host to support their germline propagation [54]. In doing so, they copied the associated TF binding site and seeded it at the site of insertion. Transposable elements having binding sites for TFs that are active in the germline, are more likely to hijack these and transpose. Accordingly, HIF is activated in early development, when DNA methylation levels are also low [53, 55]; ancestral cooption of HIF binding sites by cryptic transcripts to increase their expression is thus plausible. In line with specific TFs preferentially acting on particular retrotransposon subfamilies, we observe enrichment of HIF binding and activation at LTRs, particularly at the LTR of ERVKs.

Finally, we uncover an intriguing opportunity for cancer immunotherapy. Chiapinelli et al. already demonstrated that aza-induced cryptic transcripts are highly immunogenic and can sensitize tumors to checkpoint immunotherapy [33], while Sheng et al. showed that also histone demethylase LSD1-ablation increases cryptic transcripts, thereby enabling checkpoint blockade [56]. The mechanism underlying immunogenicity likely depends on the formation of dsRNA, which via a viral mimicry-mediated process activates the immune system [35, 48, 56]. In addition, some of these transcripts contain open-reading frames, which could translate into abnormal proteins that can be antigenic [48]. Importantly, hypoxia is endemic to most solid tumors, and therefore could have a more widespread impact than aza. Indeed, in hypoxic tumors with high checkpoint expression, DNA methylation at TSS of cryptic transcripts was reduced and consequently, cryptic transcript expression increased. Since tumors with high checkpoint expression often respond to checkpoint immunotherapy, and as cryptic transcripts could sensitize tumors to checkpoint blockade [33, 35], this suggests hypoxia-induced cryptic transcripts to play an important role in mediating the therapeutic effects exerted by checkpoint inhibitors. In contrast, immune-cold tumors characterized by low immune checkpoint expression were much less tolerant to cryptic transcript expression, showing high methylation at retrotransposon promoters. In light of our findings that methylation directly repels HIF binding, this suggests DNA methylation to block hypoxia-induced cryptic transcript expression in immune-cold tumors to maintain immunotolerance. Pharmacological demethylation of immune-cold 4T1 tumors indeed increased cryptic transcription, enhanced immunogenicity, and reduced tumor growth

in a HIF-dependent manner. By showing that low-immunogenic, hypoxic tumors can be rendered immunogenic through DNA methylation inhibitors, we thus highlight a novel treatment strategy for tumors otherwise refractory to immunotherapies.

## Methods

### Materials

All materials were molecular biology grade. Unless noted otherwise, all were from Sigma (Diegem, Belgium).

### Cell lines

MCF7, RCC4, SK-MEL-28, A549, 4T1, MC38, and CT26 cell lines were obtained from the American Type Culture Collection, and their identity was not further authenticated. None of these cell lines are listed in the database of commonly misidentified cell lines maintained by ICLAC. MCF7 *HIF1B*-knockout cells were previously described [20]. MCF7, RCC4, A549, MC38, and 4T1 cells were cultured at 37 °C in Dulbecco's modified Eagle's medium (DMEM) with 10% fetal bovine serum (FBS), 5 mL of 100 U/mL Penicillin-Streptomycin (Pen-Strep, Life Technologies), and 5 mL of L-glutamine 200 mM. SK-MEL-28 and CT26 cell lines were cultured at 37 °C in Roswell Park Memorial Institute 1640 Medium (RPMI) with 10% FBS 1% Pen-Strep and 1% L-glutamine.

Murine embryonic stem cells (mESCs) that were triple-knockout for *Dnmt1*, *Dnmt3a*, and *Dnmt3b* (*Dnmt*-TKO) and triple-knockout for *Tet1*, *Tet2*, and *Tet3* (*Tet*-TKO) and their appropriate wild-type (WT) control mESCs were obtained from Dr. Masaki Okano and Dr. Guoliang Xu respectively [57, 58]. mESCs that were knockout for *Hif1b* (*Hif1b-KO)* and their WT control mESCs were previously described [21]. *Dnmt*-TKO, *Tet*-TKO, *Tet*-WT, *Hif1b*-WT, and *Hif1b*-KO mESCs were cultured feeder-free in fibroblast-conditioned medium (DMEM with 4500 mg /L glucose, 2 mM L-glutamine, 1 mM sodium pyruvate, 15% FBS, 1% Pen-Strep, 0.1 mM of non-essential amino acids, 0.1 mM β-mercaptoethanol) on 0.1% gelatine-coated plates. mESCs from the 159 background used for the recombinase-mediated cassette exchange reaction were kindly provided by Prof. Dirk Schubeler (Friedrich Miescher Institute for Biomedical Research, Basel, Switzerland) and grown in ESC medium (DMEM with 4500 mg /L glucose, 2 mM L-glutamine, 1 mM sodium pyruvate, 15% FBS, 1% Pen-Strep, 0.1 mM of non-essential amino acids, 0.1 mM β-mercaptoethanol, $10^3$ U LIF ESGRO (Millipore)) containing 25 μg/mL hygromycin (50 μl of 5 mg/mL stock per 10 mL medium) for at least 10 days. 4T1 cells that were knockout for *Hif1b* (*Hif1b*-KO) and their WT control cells were cultured at 37 °C in DMEM with 10% FBS, 5 mL of 100 U/mL Pen-Strep, 10 μg/mL of blasticidin (ant-bl-05, Invivogen), puromycin (P9620, Sigma-Aldrich) 1.5 μg/mL medium, and 5 mL of L-glutamine 200 mM.

All cell cultures were confirmed to be mycoplasma-free every month.

### Cell line treatment conditions

Cell cultures were grown under atmospheric (21%) oxygen concentrations in the presence of 5% $CO_2$, or rendered hypoxic by incubating them under 0.5% oxygen (5% $CO_2$ and 94.5% $N_2$). For ChIP-seq experiments, hypoxia was induced during 16 h, whereas 24 h of exposure were used when assessing effects of hypoxia on gene or protein

expression level. Where indicated, cells were pre-treated with 5-aza-2′-deoxycytidine (aza, 1 μM) for 3 days by adding the required volume to fresh culture medium. Equal volumes of the carrier (DMSO) were used as control. This was followed by another day of exposure to aza in hypoxia or normoxia, bringing the total aza exposure time for experiments to 4 days. Then, 2 mM of DMOG (dimethyloxalylglycine, Sigma) was added to culture medium for 24 h where indicated. Cytotoxicity was tested using sulforhodamine B assays as described [59]. Cells were always plated at a density tailored to reach 80–95% confluence at the end of the treatment. Fresh medium was added to the cells just prior to hypoxia. To prove that the extent to which cells were exposed to hypoxia was similar across experiments, we assessed that induction of hypoxia marker genes (*BNIP3, EGLN, ALDOA, CA9*) but not *HIF1A* occurred in each experiment (Additional file 1: Fig. S2). For experiments involving exposure to aza, we assessed the expression of cancer testis antigens as a positive control (Additional file 1: Fig. S2).

### LC-ESI-MS/MS of DNA to measure 5mC
DNA was extracted and processed for LC-ESI-MS/MS to determine 5mC concentrations exactly as described previously [1].

### Western blot
To assess HIF1α protein stabilization, proteins were extracted from cultured cells as follows: cells were placed on ice, washed twice with ice-cold PBS, and lysed in protein extraction buffer (50 mM Tris-HCl, 150 mM NaCl, 1% Triton X-100, 0.5% Na-deoxycholate, 0.1% SDS, and 1× protease inhibitor cocktail (Roche)). Protein concentrations were determined using a bicinchoninic acid protein assay (BCA, Thermo Scientific) following the manufacturer's protocol. An estimated 60 μg of protein was loaded per well on a NuPAGE Novex 3–8% Tris-Acetate Protein gel (Life Technologies), separated by electrophoresis and blotted on polyvinylidene fluoride membranes. Membranes were activated with methanol and washed with Tris-buffered saline (TBS; 50 mM Tris-HCl, 150 mM NaCl) with 0.1% Tween 20, and incubated with rabbit α-tubulin (2144S, Cell Signaling), rabbit β-actin (4967, Cell Signaling), rabbit HIF-1β/ARNT (D28F3) XP® (5537, Cell Signaling) at 1:1000 dilution, and rabbit HIF-1α (C-Term) Polyclonal Antibody (Cayman Chemical Item 10006421) 1:3000. Incubation with the secondary antibodies and detection were performed according to routine laboratory practices. Western blotting was done on 3 independent biological replicates.

### Analysis of HIF1β target genes using ChIP-seq
$20–25 \times 10^6$ cells were incubated in hypoxic conditions for 16 h. Cultured cells were subsequently immediately fixed by adding 1% formaldehyde (16% formaldehyde (w/v), methanol-free, Thermo Scientific) directly in the medium and incubating for 8 min on a flat-bed shaker at room temperature (RT). Fixed cells were incubated with 150 mM of glycine for 5 min to revert the cross-links, washed twice with ice-cold PBS 0.5% Triton-X100, scraped, and collected by centrifugation ($1000 \times g$, 5 min at 4 °C). The pellet was resuspended in 1400 μL of RIPA buffer (50 mM Tris-HCl pH 8, 150 mM NaCl, 2 mM EDTA pH 8, 1% Triton-X100, 0.5% Sodium deoxycholate, 1% SDS, 1% protease inhibitor) and transferred to a new Eppendorf tube. The lysate was homogenized by passing through an insulin syringe and incubated on ice for 10 min. The chromatin was

sonicated for 3 min by using a Branson 250 Digital Sonifier with 0.7 s "On" and 1.3 s "Off" pulses at 40% power amplitude, yielding predominantly fragment sizes between 100 and 500 bps. The sample was kept ice-cold at all times during the sonication. Next, samples were centrifuged (10 min at 16,000 × G at 4 °C) and supernatant transferred in a new Eppendorf tube. Protein concentration was assessed using a BCA. A total of 50 μL of sheared chromatin was used as "input," and 1.4 μg of primary ARNT/HIF1β monoclonal antibody (NB100C124, Novus) per 1 mg of protein was added to the remainder of the chromatin and incubated overnight at 4 °C in a rotator. Next, Pierce Protein A/G Magnetic Beads (Life Technologies) were added to the samples in a volume that is 4× the volume of the primary antibody and incubated at 4 °C for at least 5 h. A/G Magnetic Beads were collected and washed 5 times with washing buffer (50 mM Tris-HCl, 200 mM LiCl, 2 mM EDTA, pH 8, 1% Triton, 0.5% sodium deoxycholate, 0.1% SDS, 1% protease inhibitor), and twice with TE buffer. The A/G magnetic beads were resuspended in 50 μL of TE buffer, and 1.5 μL of RNAse A (200 units, NEB, Ipswich, MA, USA) was added to the A/G beads samples and to the input, incubated for 30 min at 37 °C. After addition of 1.5 μL of Proteinase K (200 units, NEB) and overnight incubation at 65 °C on a stirrer, the beads were removed from the solution using a magnet and DNA was purified using 1.8× volume of Agencourt AMPure XP (Beckman Coulter) according to the manufacturer's instructions. DNA was eluted in 20 μL of TE buffer. The input DNA was quantified on Nano-Drop. Next, 1 μg of the input and all the immunoprecipitated DNA was converted into sequencing libraries using the NEBNext DNA library prep master mix set (NEB) following manufacturer's instructions.

A single end of these libraries was sequenced for 50 bases on a HiSeq, either HiSeq2500 or HiSeq4000 (Illumina), mapped using Bowtie and extended for the average insert size (250 bases). ChIP peaks were called by Model-based Analysis for ChIP-Seq (MACS) [16], with standard settings and using read counts from an input sample as baseline.

HIF1β binding peak positions in the human cell lines MCF7 (both vehicle- and aza-treated), RCC4, A549, and SK-MEL-28 were defined by using the stringent threshold $P < 10^{-15}$. A threshold equal to $P < 10^{-10}$ was used to define HIF1β binding peaks in murine cell lines (4T1, *Dnmt*-WT, and *Dnmt*-TKO ESCs).

To compare HIF1β binding peaks between human cell lines (MCF7, RCC4, A549, and SK-MEL-28), HIF1β binding peaks were called as present if the average coverage at the 200 bps centered on the summit was > 4-fold bigger than the local background, and as absent if it was < 2.5-fold smaller than the local background, with local background being defined as the read depth across regions 1.5–5 kb up- and downstream of the peak. Intermediate coverage was annotated as unclassified. To compare HIF1β binding peaks between murine *Dnmt*-WT and *Dnmt*-TKO ESCs, the HIF1β binding peak was called as present if the average coverage at the 200 bps centered on the summit was > 4-fold bigger than the background, and as absent if it was < 4-fold smaller than the background. To compare efficiency between experiments, scatter plots of read counts at peak regions of HIF1β binding regions were generated per cell line in a pairwise fashion.

## Annotation of genomic features

Human sequences were mapped to genome build hg19 and murine sequences to genome build mm10. Putative HIF binding sites were detected genome-wide by screening

the whole genome for RCGTG motifs using the regular expression search tool dreg (www.bioinformatics.nl/cgi-bin/emboss/help/dreg). The frequency per bp of RCGTG motifs inside HIF1β binding peaks and in the rest of the genome was calculated, and enrichment of RCGTG motifs at HIF1β binding peaks quantified by overlapping RCGTG positions in the genome with the HIF1β binding peak positions as defined by MACS.

The distances of HIF1β peaks to the nearest RCGTG motif (cumulative frequency), TSS, and open chromatin (frequency) were calculated by overlapping each genomic feature with HIF1β peak positions using BedTools in R. Protein-coding genes were annotated as per Ensembl version 92. Promoter regions were annotated as being 2 kb upstream or 500 bp downstream of the start site of each gene. Chromatin state annotation of MCF7 and mESCs was as described [23, 60]. HIF1β binding peaks were annotated with these features and overlapped with the repeat genome using BedTools. To assess enrichment of HIF binding at repeats, HIF1β binding peaks were 10,000 times either randomly distributed throughout the genome, or randomly distributed while matching the distal binding peak distribution. Next, the frequency of repeat binding in a random distribution was compared to that in the observed distribution. Peaks randomly assigned to poorly mapping regions were discarded.

### Genome distribution of 5mC: BS-seq, SeqCapEpi BS-seq and mDIP-seq

BS-seq, SeqCapEpi BS-seq, and mDIP-seq were performed as described previously [1]. To quantify DNA methylation inside HIF1β binding peaks, SeqCapEpi probes with > 40× coverage were overlapped with HIF1β binding peaks as defined by MACS. Methylation levels at the probes overlapping and non-overlapping (rest of the genome) HIF1β binding peaks were calculated using Seqmonk.

ChIP-BS-seq was done as ChIP-seq, except that methylated adaptors (NEB) were ligated, and DNA libraries were bisulfite-converted using the EZ DNA Methylation-Lightning™ kit (Zymo) prior to library amplification using HiFi Uracil+ (KAPA). Reads were mapped using Bismark as described [1].

### RNA-seq

To assess the impact of HIF binding at gene promoters on their expression, strand-specific RNA-seq was performed in human cell lines and murine *Dnmt*-WT and *Dnmt*-TKO ESCs. Briefly, total RNA was extracted using TRIzol (Invitrogen), and remaining DNA contaminants in 17–20 μg of RNA were removed using Turbo DNase (Ambion) according to the manufacturer's instructions. RNA was repurified using the RNeasy Mini Kit (Qiagen). For total RNA-seq, ribosomal RNA present was depleted from 5 μg of total RNA using the RiboMinus Eukaryote System (Life technologies). cDNA synthesis was performed using the SuperScript® III Reverse Transcriptase kit (Invitrogen). Three micrograms of random Primers (Invitrogen), 8 μL of 5× first-strand buffer, and 10 μL of RNA mix were incubated at 94 °C for 3 min and then at 4 °C for 1 min. Next, 2 μL of 10 mM dNTP Mix (Invitrogen), 4 μL of 0.1 M DTT, 2 μL of SUPERase• In™ RNase Inhibitor 20 U/μL (Ambion), 2 μL of SuperScript™ III RT (200 units/μL), and 8 μL of Actinomycin D (1 μg/μL) were added, and the mix was incubated 5 min at 25 °C, 60 min at 50 °C, and 15 min at 70 °C to heat-inactivate the reaction. The cDNA was purified using 80 μL (2× volume) of Agencourt AMPure XP and eluted in 50 μL of

the following mix: 5 μL of 10× NEBuffer 2, 1.5 μL of 10 mM dNTP mix (10 mM dATP, dCTP, dGTP, dUTP, Sigma), 0.1 μL of RNaseH (10 U/μL, Ambion), 2.5 μL of DNA Polymerase I Klenov (10 U/μL, NEB), and water until 50 μL. The eluted cDNA was incubated for 30 min at 16 °C, purified by Agencourt AMPure XP, and eluted in 30 μL of dA-Tailing mix (2 μL of Klenow Fragment, 3 μL of 10× NEBNext dA-Tailing Reaction Buffer, and 25 μL of water). After 30 min incubation at 37 °C, the DNA was purified by Agencourt AMPure XP, eluted in TE buffer, and quantified on NanoDrop. Subsequent library preparation was done using the DNA library prep master mix set, and sequencing was performed as described for ChIP-seq.

mRNA capture and stranded library preparation of RNA from MCF7 cells, mouse cell lines, and tumors for the purpose of retrotransposon and cryptic transcript expression analysis was performed using the KAPA Stranded mRNA-Seq Kit according to the provided protocol (Illumina). For expression analysis of coding genes, RNA-seq reads were mapped to the human or murine genome reference (hg19 or mm10) using Tophat2. Gene read numbers were counted using HTSeq and normalized to the sum of the mapped expression counts. Gene expression was presented as transcript per million (TPM), 0.01 offset. Differential gene expression was tested using edgeR.

Expression of cancer testis antigens was annotated according to all entries listed in the CTDatabase (www.cta.lncc.br/modelo.php). Cytolytic activity was quantified as the log average (geometric mean in TPM) of the RNA expression of 2 key cytolytic enzymes: granzyme A (GZMA) and perforin 1 (PRF1).

### RepEnrich analyses

RNA-seq data were expressed in TPM with an offset of 0.01. Expression read counts of retrotransposons are calculated using the RepEnrich tool (https://github.com/neretti-lab/RepEnrich) and normalized to the total mappable read depth. The repeat genome of the human reference genome hg19 was downloaded from the RepEnrich website. Human retrotransposon classes (LINE, SINE, LTR) contain 16 families and 779 subfamilies. The repeat genome of the mouse genome mm10 was built using the Repeat-Masker track from the UCSC genome browser. Mouse retrotransposon classes (LINE, SINE, LTR) contain 24 families and 906 subfamilies.

### CREDENToR analysis

The overall strategy of CREDENToR is to perform de novo assembly of all reads and based on this define all cryptic transcripts. CREDENToR will consider transcripts encompassing more than one repeat element as one cryptic transcript and quantify gene expression for each of them. To achieve this, fastq files of RNA-seq data were first aligned to the human (build GRCh38) or the murine genome (build mm10) using STAR (version 2.5) with a tolerance of two mismatches. Transcriptomes were subsequently assembled using StringTie [61] (version 1.3.4d), under guidance of the transcript annotation tool Ensembl 92. All de novo assembled transcription annotations from the same set of tumor samples (i.e., MCF7 or 4T1 cell lines, or each of the 14 tumor types downloaded from TCGA) were merged using "StringTie --merge". HTSeq-counts [62] (version 0.11.2) were used to count the read numbers of known and novel genes. Noncoding transcripts (transcripts not overlapping annotated coding genes) in

the merged transcription annotations were assigned as cryptic transcripts when any of their exons overlapped with a retrotransposon repeat annotation (LTR, LINE, or SINE, based on RepeatMasker annotation from UCSC). If a transcript overlapped with > 1 annotated repeat, the retrotransposon with the highest overlap was assigned to this cryptic transcript.

For the analysis of MCF7 data, the assembled annotations from all experimental conditions involving MCF7 cells assessed in vitro were merged before read counting. For the analysis of 4T1 data, the assembled annotations from in vitro and in vivo samples were merged together. Cryptic transcripts were considered to be HIF-associated if a HIF binding summit was detected within the transcript promoter (i.e., 2000 bp upstream and 500 bp downstream of the transcription start site). Per set of experiments (24 samples), we further required that the read number per cryptic transcript exceeds 10 in at least 1 sample and that the reads per kb per million reads (RPKM) exceed 1 in at least one sample. For these cryptic transcripts, DESeq (version 1.30.0) was used to test the differences between each pair of conditions. For the TCGA data, we merged assembled transcription annotations for each tumor type separately. Cryptic transcripts were calculated using the total cryptic transcript read count divided by the total coding gene read count. In volcano plots, individual cryptic transcripts were plotted, but in violin plots, where we compare effects to the cryptic transcripts obtained by RepEnrich, we summed cryptic transcript counts into retrotransposon subfamilies, log-transferred counts-per-million (normalized to total read counts), and considered those as expression values. Violin plots invariably show > 95% of data points. *P* values were corrected for multiple testing following Benjamini and Hochberg correction. The CREDENToR pipeline has been made available on GitHub (https://github.com/Jieyi-DiLaKULeuven/CREDENToR).

### Gene ontology analysis

Genes were associated to ontologies as annotated in BioMART (Ensembl GRCh37 release 84), and enrichment of ontologies was analyzed using TopGo (version 1.0) in R [63], using the *classic* algorithm, contrasting to all protein-coding genes.

### Structural modeling of DNA methylation

The crystal structure of HIF2α:HIF1β in complex with DNA containing the RCGTG core sequence 5′-ACGTG-3′ ([25], PDB code 4ZPK) was used as a template for introducing and analyzing the structural consequences of methyl groups at position 5 of the cytosines using the programs PyMOL (Schrodinger, LLC) and Chimera [64].

### Microscale thermophoresis (MST) binding assay

MST measurements were performed in triplicate using the NanoTemper Monolith NT.115 instrument. The two protein complexes (HIF1α-HIF1β and HIF2α-HIF1β) were purified as described earlier [25]. They were both labeled using Monolith NT Protein labeling kit RED-NHS (Nano Temper technologies). Oligonucleotides were from IDT. In brief, 25 nM of each labeled protein was mixed in 16 serial dilutions of 1:1 with different DNA concentrations starting from a concentration of 25 μM. The experiment was carried out in 20 mM phosphate buffer, 75 mM NaCl, 5 mM DTT, 0.05% Tween-20, and pH 7.4. Samples were incubated for 20 min on ice prior to loading 5 μL of each sample into the standard treated capillaries. MST measurements were carried out at

25 °C at 20% LED power and medium MST power. Data was normalized to % fraction bound, and the values for the equilibrium dissociation constant ($K_D$) were calculated by fitting the curves in GraphPad Prism 7.

### Generation of mESCs containing a methylated or unmethylated HIF binding region

A DNA fragment (human chr16:30,065,212-30,065,711) containing five CGTG motives was selected based on high HIF1β ChIP-enrichment in MCF7, RCC4, and SK-MEL-28 cells. Oligonucleotides were designed to amplify the target region (AGGTGCAATT GTTCCTCCGCCTCCCTTAC and AAGGGCAATTGCCGAGCTTTTTCCTTTACGA) and used for PCR amplification of the target region using the Q5® Hot Start High-Fidelity 2× Master Mix (NEB), followed by evaluation of the PCR products by gel electrophoresis and purification with the Qiaquick PCR purification kit (28104, Qiagen). These PCR primers were evaluated for specificity in human (MCF7, RCC4, SK-MEL-28) but not in mouse genomic DNA, while MfeI restriction sites were added to the ends of the primer pairs. The purified amplicon was digested with MfeI and cloned into the L1-poly-L1 plasmid (provided by Prof. Dirk Schubeler, Friedrich Miescher Institute for Biomedical Research, Basel, Switzerland), containing a multiple cloning site flanked by two inverted L1 *Lox* sites. Correct insertion and sequence identity were verified by Sanger sequencing. This plasmid was in vitro methylated using M.SssI (NEB) according to the manufacturer's instructions and purified using isopropanol precipitation. Successful and complete in vitro methylation was confirmed by bisulfite-conversion (EZ DNA Methylation-Lightning Kit, D5031, Laborimpex), PCR amplification using the MegaMix Gold 2× Mastermix (Microzone), and Sanger sequencing. Ten micrograms of pIC-CRE plasmid and 25 μg of (un)methylated plasmid were electroporated in murine ES 159 cells containing an L1-flanked thymidine kinase expression cassette (provided by Prof. Dirk Schubeler, Friedrich Miescher Institute for Biomedical Research, Basel, Switzerland). After electroporation, cells were plated and maintained in nonselective ES medium for 1 day and, from the second day onwards, cultured in ES medium containing 10 μM ganciclovir. After 10 to 12 days, individual clones of the surviving cells were picked and transferred to ES medium in 96-well plates, then gradually expanded and, following DNA extraction, assessed for occurrence of successful insertion events using PCR (using the following oligonucleotides: AGGTGCAATTGTTCCTCCGCCTCCCTTAC   and   AAGGGCAATTGCCGAGCT TTTTCCTTTACGA) and gel electrophoresis.

To verify maintenance of the methylation levels of the cloned HIF binding site, genomic DNA was extracted from a positive clone. Then, 500 ng of DNA was bisulfite-converted using the EZ DNA Methylation-Lightning Kit (D5031, Laborimpex) and amplified using the MegaMix Gold 2× Mastermix and validated primer pairs for the target locus (Forward: GTTTGGGTTAGTGATAGGGTGT, Reverse: AAACCCTCCC TTCTACTCCTTTCC). Per sample, PCR product sizes were verified by gel electrophoresis, and amplicons converted into sequencing libraries using the NEBNext DNA library prep master mix set (E6040L, Bioke). These were next sequenced to a depth exceeding 500×, and mapped and analyzed as described higher.

Positive colonies were expanded into 10-cm dishes and subjected to ChIP as described above. qPCR was performed with the SYBR GreenER™ qPCR SuperMix Universal (11762500, Life Technologies) on a Quantstudio 12K (Applied Biosystems), by using

specific primers for the cloned locus (oligonucleotides TCGTTTCCGACTTTTCCATC and CAGCCAGAATGTTGGCAAT) and an independent murine genomic region for background quantification (oligonucleotides CACTTGCTGAATAATTGGGTGT and CTGTTGTCCAGTTTTCTTCACG). Enrichment was calculated as fold enrichment over background.

### TCGA samples and data analysis

From the TCGA server, we selected 5193 tumors from 14 tumor types: 413 bladder urothelial carcinoma (BLCA), 664 breast cancer (BRCA), 303 cervical squamous cell carcinoma and endocervical adenocarcinoma (CESC), 201 colon adenocarcinoma (COAD), 497 head and neck squamous cell carcinoma (HNSC), 269 kidney renal papillary cell carcinoma (KIRP), 372 liver hepatocellular carcinoma (LIHC), 460 lung adenocarcinoma (LUAD), 368 lung squamous cell carcinoma (LUSC), 175 pancreatic adenocarcinoma (PAAD), 498 prostate adenocarcinoma (PRAD), 465 skin cutaneous melanoma (SKCM), 338 stomach adenocarcinoma (STAD), and 171 uterine corpus endometrial carcinoma (UCEC) for which RNA data were available. The corresponding RNA-seq read counts were downloaded. DNA methylation data from Infinium Human-Methylation450 BeadChip arrays available were downloaded for the same samples.

Tumor types were classified as responsive or non-responsive to checkpoint immunotherapy following the classification described by Turajlik and colleagues [38], with three notable exceptions. Firstly, kidney renal papillary cell carcinoma (KIRP) was classified as non-responsive as no study has yet demonstrated responsiveness of this tumor type. Secondly, clear cell kidney carcinoma (KIRC) tumors were not analyzed, as the HIF-constitutive activation in these tumors (due to VHL-loss) precludes their classification into a hypoxic and normoxic subset. Finally, also microsatellite-instable COAD tumors were discarded from all analyses, as these tend to show constitutive hypermethylation, precluding their stratification in high and low methylation subgroups [65]. Importantly, the tumor types that we define as responsive will still contain tumors that fail to respond, whereas the non-responsive tumor types will also contain a minority of tumors that do respond to immunotherapy. For instance, a subset of triple-negative breast tumors responds to checkpoint immunotherapy. Likewise, there is evidence that a small subset of LIHC tumors also responds and that microsatellite-instable tumors also occur in UCEC tumors. Cryptic transcription loads were calculated using the total cryptic transcript read count divided by the total coding gene read count.

For the methylation stratification, the beta values of HM450K methylation microarray data were downloaded from TCGA. Probes overlapping the cryptic transcript promoter (i.e., 2000 bp upstream and 500 bp downstream of the transcription start site) were regarded as cryptic transcription-associated probes. For each tumor, its promoter methylation level was calculated as the mean beta value of all cryptic transcription-associated probes. All tumors were then classified into high and low methylation groups based on the median value of methylation levels.

To identify which of these tumor samples were hypoxic or normoxic, we performed unsupervised hierarchical clustering based on a modification of the Ward error sum of squares hierarchical clustering method (Ward.D of the clusth function in R's stats package) on normalized log-transformed RNA-seq read counts for 15 genes that make up

the hypoxia metagene signature (*ALDOA, MIF, TUBB6, P4HA1, SLC2A1, PGAM1, ENO1, LDHA, CDKN3, TPI1, NDRG1, VEGFA, ACOT7, CDKN3*, and *ADM*) [37]. In each case, the top 2 subclusters identified were annotated as normoxic and hypoxic.

To test the interaction between hypoxia and DNA methylation, we assessed read counts for cryptic transcripts in two negative-binominal generalized linear models with both oxygenation (hypoxic and normoxic; encoded as 0 and 1) and methylation (low and high methylation; encoded as 0 or 1), with or without an interaction term. Both models were compared to each other using DESeq. A positive interaction coefficient represents a cooperative enhancement of cryptic transcript expression in low-methylation, hypoxic tumors. To further enrich for tumors that are prone to respond to checkpoint immunotherapy, we stratified all tumor types into high *PDL1* mRNA expressing and low *PDL1* mRNA expressing tumors, and into tumors with a high or low tumor mutation burden (TMB). Stratification was done on the third decile in both cases. TMB was estimated based on the number of substitutions identified by TCGA in each tumor sample. All substitutions were considered, except for those also present in non-malignant samples (i.e., exclusion of germline variants) or those clustering within and across different samples (and therefore most likely representing sequencing or mapping errors).

### Single-cell analysis

We used CREDENToR to map cryptic transcript expression in each individual cell from a public single-cell RNA-seq dataset [66]. The cryptic transcript annotation was obtained from the CREDENToR analysis of lung TCGA tumors (LUAD and LUSC). CellRanger (version 1.1.0) was used as the mapping tool. The annotation of each individual cell is as previously defined in Lambrechts et al. [66].

To study the effect of hypoxia and DNA methylation on cryptic transcript expression, we used a public single-cell dataset GSE97693 [39]. Single-cell RNA-seq reads were downloaded and mapped using STAR (version 2.5). Cancer cells for which the number of uniquely mapped reads exceeded 30% were stratified into hypoxic cells and normoxic cells as described higher. The cryptic transcript annotation was obtained from COAD tumors in TCGA. We selected 458 cryptic transcripts expressed in at least 20% of cells. Methylation was defined as the number of methylated CpGs over the total number of CpGs in a region 2 kb downstream and 500 bp upstream of the cryptic transcript transcription start site.

### 4T1 *Hif1b*-knockout

Four gRNAs targeting two different exons in the *Hif1b* locus of the mouse genome and one non-targeting gRNA (scramble) were designed with the appropriate restriction sites for the receiver plasmid using the online Crispor tool (http://crispor.tefor.net). Oligonucleotides corresponding to gRNAs were synthesized by IDT, and forward and reverse oligonucleotides were annealed in the CutSmart buffer (B7240S, NEB) before cloning into the LentiGuide-Puro plasmid (Plasmid 52963, Addgene). Positive colonies were screened by PCR and validated by Sanger sequencing. LentiGuide-Puro plasmids containing GFP were used as positive control to evaluate the transfection and transduction efficacy.

A transformation mix containing viral particles, TE, CaCl$_2$, H$_2$O, and LentiGuide-Puro plasmid was added to HEK 293T cells when reaching 70% confluency. Four plasmids containing the different gRNAs for *Hif1b* and one plasmid containing the scramble gRNA were used, together with plasmids containing GFP as positive control. Medium was renewed after 14–16 h, and transfection efficiency was evaluated based on GFP expression. After 36 h, supernatant containing the concentrated virus was collected by ultracentrifugation. Virus was dissolved in clean PBS and stored at − 150 °C.

4T1 cells were transduced with a lentiviral vector expressing a doxycycline-inducible Cas9 nuclease (Cat # CAS11229, Dharmacon) for a tight regulation of the Cas9 expression and gene editing. An infection rate of 30% was used to ensure that the majority of transduced cells harbor a single copy of the vector. These 4T1 cells were always kept in selection medium containing 10 μg/mL of blasticidin (ant-bl-05, Invivogen). When reaching 70% confluency, cells were transduced with one titer of virus. After 24 h, the virus was removed and transduction efficacy evaluated based on GFP expression. After 48 h, puromycin (P9620, Sigma-Aldrich) 1.5 μg/mL medium was added to the medium. Cells were kept in the presence of blasticidin and puromycin for the remaining experimental procedures. After 3–5 days, Cas9 expression was induced by adding doxorubicin (D2975000, Sigma-Aldrich) 0.5 μg/mL medium for 3 days. Cells were kept 1 day without doxorubicin before injection in the mice or further experimental procedures. 4T1 cells transduced with the four gRNAs targeting *Hif1b* were expanded, and proteins were extracted to test the efficacy of the knockout by Western blot. The most efficient gRNA was used to perform the further experiments (F: CACCGTGAAATAGAACGG CGGCGA and R: AAACTCGCCGCCGTTCTATTTCAC; non-targeting F: CACCGC ACTACCAGAGCTAACTCAG and non-targeting R: AAACCTGAGTTAGCTCTG GTAGTGC). Stability of knockout in these polyclonal 4T1 cells after 2 weeks was confirmed by Western blot.

**Mouse tumor model**

All the experimental procedures were approved by the Institutional Animal Care and Research Advisory Committee of the KU Leuven. $1 \times 10^6$ 4T1 cells, 4T1 *Hif1b*-knockout, or wild-type 4T1 cells (scramble) were injected orthotopically in the mammary gland of 10 weeks old Balb/c mice, and $1 \times 10^6$ CT26 or MC38 cells were injected subcutaneously in 10 weeks old Balb/c or C57BL/6J mice respectively. When the tumor was palpable (starting volume 100 mm$^3$), the mice were injected intraperitoneally with 0.8 mg/kg of 5-aza-2′-deoxycytidine (aza) or PBS, 40 mg/kg DC101 antibody (BE0089, InVivoMab) or IgG (BE0060, InVivoMab), or 10 mg/kg anti-PD1 antibody (BE0146, InVivoMab) or IgG (BE0089, InVivoMab) according to the following schedules: DC101 three times per week; anti-PD1 every other day, starting when the tumor size was around 200 mm$^3$; aza was administered in 2 cycles with 2 days rest in between until the control tumors reached the endpoint. Tumor volumes were monitored every 2 to 3 days by a caliper, and mice were culled before tumor volumes exceeded 2000 mm$^3$. When over 20% of mice were culled, the experiment was terminated (all arms). In vivo experiments in 4T1, CT26, and MC38 treated with aza or anti-PD1 antibody were performed three times, with at least 6 mice per treatment group in each experiment.

### Neo-epitope burden

To assess neo-epitope burden, we mapped RNA sequencing data of isogenic 4T1, B16, and CT26 tumor models, removed duplicate reads from individual samples, and merged per tumor model all samples into a single file. In this file, variants were called according to GATK best practices, using GATK3.4. Briefly, reads were split into exon segments and sequences overhanging the non-exonic regions were hard-clipped using split'n'trim. Next, local indel realignment and base recalibration was performed, followed by variant calling with GATK's HaplotypeCaller. After quality filtering for minimal Fisher strand values (30) and minimal read depth (10-fold), we removed SNPs reported in the Sanger Mouse project (rsIDdbSNPv137). Remaining variants were annotated by Annovar (version 2.17.0), and only variants in coding regions were retained. Finally, the neo-epitope burden was expressed as the number of non-SNP variants in coding sequences, normalized to the number of coding sequences that were expressed, the latter being defined as having a minimal read depth of 10.

### Immunofluorescence analysis of tumor sections

Different protocols were applied depending on the epitope of interest: hypoxia (pimonidazole) staining was combined with blood vessel (CD31) staining, and cytotoxic T cell activity (granzyme B) with T cell infiltration (CD8a) staining. General (CD45) and cytotoxic (CD8a) T cell infiltration stainings were performed separately. Tumors were harvested, fixed in formaldehyde, and embedded in paraffin using standard procedures. Slides were deparafinated and rehydrated in 2 xylene baths (5 min), followed by 5 times 3 min in EtOH baths at decreasing concentrations (100%, 96%, 70%, 50%, and water) and a 3 min Tris-buffered saline (TBS; 50 mM Tris, 150 mM NaCl, pH 7.6) bath. Antigen retrieval was done using AgR (DAKO) at 100 °C for 20 min, followed by cooling for 20 min. Slides were washed in TBS for 5 min, and endogenous peroxidase activity was quenched using $H_2O_2$ (0.3% in MeOH), followed by three 5-min washes in TBS. Slides were blocked using pre-immune goat serum (X0907, Dako) or pre-immune rabbit serum (for pimonidazole, X090210, Dako) 20% in TNB. Binding of primary antibodies FITC-conjugated mouse anti-pimonidazole (HP2-100, Hydroxyprobe), rabbit anti-Gzmb (ab4059, Abcam), and rat anti-CD45 (553,076, BD Biosciences) all 1:100 in TNB was allowed to proceed overnight. Slides were washed 3 times in TNT (0.5% Triton-X100 in TBS) for 5 min, after which secondary antibodies peroxidase-conjugated rabbit anti-FITC (PA1-26804, Pierce), Alexa Fluor 488-conjugated goat anti-rabbit (A-11034, Thermo Fisher), and biotinylated goat anti-rat (559286, BD biosciences) all 1:100 in TNB with 10% pre-immune goat serum were allowed to bind for 1 h. Slides were washed 3 times for 5 min in TNT, after which signal amplification was done by 30 min incubation with peroxidase-conjugated streptavidin 1:100 in TNB (for all besides pimonidazole) accompanied by nuclear staining with Hoechst (H3570, Thermo Fisher) 1:500 in TNB only for the single (CD45 or CD8a) stainings, washing (3 times 5 min in TNT), and 8 min incubation using fluorescein tyramide (for pimonidazole NEL701A001KT, Perkin Elmer) or Cy3 (NEL704A001KT, Perkin Elmer) 1:50 in amplification diluent.

Slides stained for pimonidazole and Gzmb required co-staining for CD31 and CD8a respectively and were subjected to a second indirect staining for the latter epitopes. After 5 min of TNT and 5 min of TBS, slides were quenched again for peroxidase

activity using $H_2O_2$ and blocked using pre-immune goat or rabbit (CD31) serum, prior to a second overnight round of primary antibody binding: rat anti-CD31 (557355, BD Biosciences) or rat anti-CD8a (14-0808-82, Thermo Fisher) 1:100 in TNB. The next day, 3 times 5 min washes with TNT were followed by a 1 h incubation with biotinylated goat anti-rat (559286, BD biosciences) 1:100 in TNB, again 3 times 5 min washes with TNT, a 30-min incubation with peroxidase-conjugated streptavidine 1:100 in TNB accompanied by nuclear staining with Hoechst (H3570, Thermo Fisher) 1:500 in TNB, 3 times 5 min washes with TNT and signal amplification for 8 min using Cy3 (NEL704A001KT, Perkin Elmer) 1:50 in amplification diluent. Finally, slides were washed 3 times for 5 min with TNT and mounted with Prolong Gold (P36930, Life Technologies).

For immunofluorescence analysis on 4T1 wild-type tumors, slides were imaged on an infraMouse Leica DM5500 microscope. Four sections from different treatment groups were stained per slide while 6 pictures from different tumor areas were used for processing with ImageJ. More specifically, nuclei were identified using the Hoechst signal, and signal intensities for fluorescein (pimonidazole), Alexa Fluor 488 (Gzmb), and Cy3 (CD45, CD8a and CD31) were used to detect Gzmb$^+$, CD45$^+$, and/or CD8a$^+$ cells. Analyses were exclusively performed on slide regions showing a regular density and shape of nuclei, in order to avoid inclusion of acellular or necrotic areas. Gzmb$^+$ CD8a$^+$ cells were counted directly, allowing the precise quantification of the number of active cytotoxic T cells per tumor. The number of CD45$^+$ cells was used to normalize the number of CD8a$^+$ cells, as such calculating the number of infiltrating cytotoxic T cells compared to the total immune infiltration. CD31-positive regions were quantified manually using ImageJ. The pimonidazole signal was used together with the Hoechst signal to quantify the percentage of hypoxia per tumor area in each picture and stratify tumors as hypoxic (pimo-high) or normoxic (pimo-low).

For immunohistofluorescence on 4T1 *Hif1b*-knockout or *Hif1b*-scramble grafts, tumors were harvested and snap frozen in liquid nitrogen before temporary storage at − 80 °C. Thawed tumors were embedded in paraffin and sectioned using standard procedures (5 μm of thickness). In a Leica Autostainer (30 min), slides were deparafinated and rehydrated in 2 xylene baths for 5 min, followed by 5 min in ethanol baths at decreasing concentrations (100%, 96%, 70%, 50%, and water). Slides were fixed in 10% neutral buffered formalin for 10 min and rinsed twice in double-distilled water. Antigen retrieval proceeded in AR6 buffer (AR600, PerkinElmer) at 100 °C for 23 min in a pressure cooker, followed by cooling in double-distilled water for 20 min. Slides were washed in TBST (TBS with 0.5% Tween 20) for 3 min and blocked using blocking buffer (pre-immune goat serum (X0907, Dako) 10%, 1% BSA (126575, Millipore) in TBS)) for 30 min. The primary antibody (rabbit anti-Gzmb) 1:1000 in dilution buffer (1% BSA in TBS) was applied for 30 min at RT, followed by 3 washes of 2 min in TBST at RT. Slides were next incubated with the secondary antibody (EnVision™+/HRP goat anti-rabbit (K4003, Dako)) for 10 min at RT and washed 3 times for 2 min in TBST at RT. The OPAL 570 fluorophore (fp1488, PerkinElmer) 10% in amplification diluent (FP1498, PerkinElmer) was applied for 10 min at RT followed by 3 washes of 2 min in TBST at RT. Slides were stripped by heating in AR6 buffer just below the boiling point and cooled down in double-distilled water, followed by rinsing in TBST. These steps were repeated starting from blocking for the second staining with primary antibody rat

anti-CD8a 1:300, secondary antibody goat anti-rat (MP-7444, Vector) and opal 690 (fp1497, PerkinElmer), and the third staining with rat anti-CD45 1:1000, secondary antibody goat anti-rat, and Opal 520 (fp1487, PerkinElmer). After the third staining, slides were incubated with spectral DAPI (fp1490, PerkinElmer) 10% in TBST for 5 min at RT, washed for 2 min in TBST at RT, and mounted with ProLong Diamond Antifade Mountant (P36961 Invitrogen). Images were acquired on a Zeiss Axio Scan.Z1 using a ×20 objective and ZEN 2 software (Zeiss) with exposure times between 10 and 50 ms. Image processing was done using QuPath (version 0.1.2). Specifically, following visual inspection of the staining results, cells were first automatically detected using the DAPI channel (cell size constrained between 5 and 400 $\mu m^2$). Next, a cell classifier was generated using QuPath. Specifically, for 1 slide out of all slides, 5 sets of cells were selected: one set that was positive for CD45, one set that was negative for CD45, and three sets of CD45$^+$ cells positive for CD8, Gzmb and CD8, or Gzmb alone. Using these 5 sets of cells, a random trees classifier was generated. Cell classification was visually verified to have occurred correctly. Next, in each tumor section, a representative region was selected, containing at least 1000 cells. On these cells, the random trees classifier was subsequently applied. This process was reiterated for all other tumor sections stained for the same set of markers. The resulting cell identities were then exported and processed in R. For each tumor, average cell frequencies were generated, which were summarized using boxplots.

### Immunofluorescence analysis of 4T1 cells

For the dsRNA staining on 4T1 cells, 12.000 cells were seeded on gelatin-coated glass slips in 12-well plates on day 0. After attaching for 6 h, cells were treated with aza or control (DMSO) for 3 days, with renewal of the medium after 48 h. After 72 h, the medium was refreshed and cells were incubated in hypoxia or normoxia for 16 h. After washing 3 times with PBS, cells were fixed in 1 ml of ice-cold methanol for 15 min at − 20 °C. Cells were washed 3 times with PBS and blocked for 1 h with blocking buffer (0.1% triton X-100 with 5% goat serum in PBS). Primary antibody (1:50 dilution in blocking buffer; clone J2, Scicons) was applied overnight at 4 °C. Cells were washed 3 times for 10 min with washing buffer (0.1% triton X-100 in PBS) and secondary antibody (1:500 in secondary antibody buffer (0.1% triton X-100 with 2% goat serum in PBS)). Goat anti-mouse IgG coupled to Alexa 488, Life Technologies) was applied for 1 h in the dark. Cells were washed 3 times for 10 min with washing buffer and mounted with DAPI stain on cover glasses. Slides were imaged on an infraMouse Leica DM5500 microscope. Three slides from different treatment groups were stained in triplicate (biological replicates), while 3 pictures from different slides were used for processing with ImageJ. More specifically, nuclei were identified using the DAPI signal, and nucleated cells were further selected based on particle size. Signal intensities for Alexa Fluor 488 in the selected cells were used to detect dsRNA$^+$ cells. Analyses were exclusively performed on slide regions showing a regular density and shape of nuclei, in order to avoid inclusion of acellular or necrotic areas. Mean dsRNA expression was calculated per experiment, normalized to the background signal (secondary antibody only), and expressed as mean pixel intensity relative to the control group (normoxia + DMSO).

## Supplementary information

---

**Additional file 1: Fig. S1.** HIF1β peaks in MCF7 cells under 0.5% O2. **Fig. S2.** Expression of hypoxia genes and cancer testis antigens. **Fig. S3.** Cell-type-specific HIF1β binding. **Fig. S4.** DNA methylation directly repels HIF1β binding. **Fig. S5.** Quality control of the ChIP-seq replicates. **Fig. S6.** HIF binds retrotransposons in demethylated genomes. **Fig. S7.** Examples of cryptic transcripts upregulated by HIF1β. **Fig. S8.** Cryptic transcript expression in TCGA tumors. **Fig. S9.** Aza treatment increases immunogenicity.

**Additional file 2: Table S1.** HIF1β binding peaks detected using MACS at $P < 10^{-15}$ in MCF7 cell line.

**Additional file 3: Table S2.** HIF1β binding peaks detected using MACS at $P < 10^{-15}$ across RCC4, MCF7 and SK-MEL-28 cell lines. For each cell line, HIF1β binding was annotated as 'present' if the peak area showed > 4-fold enrichment over the local read depth, and as 'absent' if it showed < 2.5-fold enrichment; intermediate enrichment scores were annotated as 'unclassified'.

**Additional file 4: Table S3.** Cryptic transcript promoter DNA methylation and expression levels. Shown are results from an analysis of combined single-cell methylome-and-transcriptome sequencing of colorectal cancer cells, as generated by Bian and colleagues [38].

**Additional file 5.** Review history.

---

### Acknowledgements

We thank Masaki Okano and Guoliang Xu for murine *Dnmt*-TKO and *Tet*-TKO ESCs and the corresponding matching WT ESCs.

### Review history

The review history is available as Additional file 5.

### Peer review information

### Authors' contributions

B.T., M.M., and D.L. contributed to the funding acquisition, supervision, and conceptualization; B.T., F.D.A, L.V.D., J.X., L.M., S.N.S., P.B-K., L.S., J.M., J. Q., R. A., M.D.B., C.S., F.R., V.C., S.K., J.D.S., and P.C. contributed to the methodology; B.T., F.D.A., J.X., and H.Z. contributed to the software; B.T., F.D.A., L.V.D., H.Z., R.V.B, L.M., S.N.S., F.D., V. C., S.K., Q.Y., and L.Z. contributed to the formal analysis; B.T., F.D.A., H.Z., L.V.D., and J.X. contributed to the investigation; B.T., W.R., S.N.S., F.D.A, and D.L. contributed to the writing of the original draft. The author(s) read and approved the final manuscript.

### Authors' information

Twitter handles: @VanDyckLaurien (Laurien Van Dyck); @RBerrens (Rebecca V. Berrens); @JunbinQian (Junbin Qian); @LiesbethMin (Liesbeth Minnoye); @MarieDeBorre (Marie De Borre); @MCSimonLab (Celeste Simon); @CarmelietLab (Peter Carmeliet); @ReikLab (Wolf Reik); @LabMazzone (Massimiliano Mazzone); @bernthie (Bernard Thienpont); @LambrechtsDlab (Diether Lambrechts).

### Funding

This work was supported by grants from the ERC (CHAMELEON 617595 to D.L.; CHAMELEO 334420 to B.T.), Stichting tegen Kanker ('Foundation contre le Cancer): FAF-C/2016/876 and FWO: G070615N. Intensive computation was provided by the Flemish Supercomputer Center (VSC). H.Z. and B.T. were supported by FWO-F postdoctoral fellowships and L.V.D by an aspirant FWO fellowship.

### Availability of data and materials

The dataset supporting the conclusions of this article is available in the Gene Expression Omnibus repository under the SuperSeries accession GSE85356, which is composed of subseries for ChIP-BS-seq (GSE85351), ChIP-seq (GSE85352), RNA-seq (GSE85353) and bisulfite-seq (GSE85354, GSE85355, and GSE152180) [67].
HIF1β, HIF1α, HIF2α, and isotype ChIP-seq data from MCF7 cells were obtained from the Gene Expression Omnibus database under accession numbers GSM700947, GSM700944, GSM700945, and GSM700948.
WGBS data were obtained from the Gene Expression Omnibus database under accession number GSM1328112 for MCF7 cells [18] and under accession number GSM1127953 for WT mESC [68].
CTCF, FOXA1, and GATA3 ChIP-seq data from MCF7 cells were obtained from the Gene Expression Omnibus database under accession numbers GSM1003581, GSM1010727, and GSE41561 [17].
NOMe-seq data were obtained from the Gene Expression Omnibus database under accession number GSE57498 [23].
Single-cell RNA-seq data for lung tumors were obtained from ArrayExpress under accessions E-MTAB-6149 and E-MTAB-6653 [66].
Single-cell RNA-seq and methylation data for colorectal tumors were obtained from the Gene Expression Omnibus database under accession number GSE97693 [39].

### Ethics approval and consent to participate

All the experimental procedures on animals were approved by the Institutional Animal Care and Research Advisory Committee of the KU Leuven.

**Competing interests**
The authors declare that they have no competing interests.

**Author details**
[1]Center for Cancer Biology, VIB, 3000 Leuven, Belgium. [2]Laboratory of Translational Genetics, Department of Human Genetics, KU Leuven, 3000 Leuven, Belgium. [3]Epigenetics Programme, Babraham Institute, Cambridge CB22 3AT, UK. [4]The Old Schools, University of Cambridge, Trinity Lane Cambridge CB2 1TN, UK. [5]Laboratory of Tumor Inflammation and Angiogenesis, Department of Oncology, KU Leuven, 3000 Leuven, Belgium. [6]Target Discovery Institute, Nuffield Department of Medicine, University of Oxford, Oxford, UK. [7]State Key Laboratory of Ophthalmology, Zhongsan Ophthalmic Center, Sun Yat-Sen University, Guangzhou, China. [8]Laboratory of Angiogenesis and Vascular Metabolism, Department of Oncology, Leuven Cancer Institute, KU Leuven, 3000 Leuven, Belgium. [9]Institute of Basic Medical Sciences, University of Oslo, 0372 Oslo, Norway. [10]Laboratory of Dermatology, Department of Oncology, KU Leuven, 3000 Leuven, Belgium. [11]Laboratory for Functional Epigenetics, Department of Human Genetics, KU Leuven, 3000 Leuven, Belgium. [12]Unit for Structural Biology, Department of Biochemistry and Microbiology, Ghent University, 9052 Ghent, Belgium. [13]VIB Center for Inflammation Research, 9052 Ghent, Belgium. [14]Abramson Family Cancer Research Institute, Perelman School of Medicine, University of Pennsylvania, Philadelphia, PA 19104, USA. [15]Department of Cell and Developmental Biology, Perelman School of Medicine, University of Pennsylvania, Philadelphia, PA 19104, USA. [16]Centre for Trophoblast Research, University of Cambridge, Cambridge CB2 3EG, UK. [17]Wellcome Trust Sanger Institute, Hinxton, Cambridge CB10 1SA, UK. [18]Department of Physiology, Development and Neuroscience, University of Cambridge, Cambridge CB2 3EG, UK. [19]Clinical and Experimental Endocrinology, Department of Chronic Diseases, Metabolism and Ageing, KU Leuven, 3000 Leuven, Belgium.

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

## 
