## [**Additional file 5.** Review history. · Genome Biology]

Review History

First round of review

Reviewer 1

Are you able to assess all statistics in the manuscript, including the appropriateness of statistical tests used? No, I do not feel adequately qualified to assess the statistics.

Comments to author:

This manuscript by Flora D' Anna et al and her colleagues has demonstrated that DNA methylation blocks the binding of hypoxia-inducible transcription factors (HIFs), and that DNA demethylation enables HIF binding to various repetitive genomic regions, inducing the expression of cryptic transcripts in a hypoxia dependent manner, thus further sensitizing tumors to checkpoint immunotherapy. This study tries to build a linkage between hypoxia, DNA hypomethylation, and tumor immunotolerance by providing experimental approves, however, the data seems to conflict with their original novel discovery (Thienpont et al. Nature, 2016).

Here are some major concerns regarding this study:

1. As mentioned before, their previous study (Thienpont et al. Nature, 2016) revealed a novel discovery in the field that tumor hypoxia caused DNA hypermethylation by reducing TET activity and was associated with increased malignancy, poor prognosis and resistance to radio- and chemotherapy. However, in this current study, the authors showed that hypoxia could induce HIF binding to the hypomethylated DNA without comparing hypermethylated sites by HIF under hypoxia situation. Therefore, under hypoxia conditions, HIF binding sites should be decreased because of DNA hypermethylation. What is the consequence of the lost HIF binding, and does this increase or decrease immunotolerance? The authors need to present data and discuss the logical approach in this study.
2. Following question 1, the experimental approach (Figure 1) is misleading, as the authors try to demonstrate the correlation between HIF1 binding sites and DNA methylation status using normoxic conditions. The best way to answer the question regarding alterations of HIF binding and DNA methylation under hypoxia, the authors should compare HIF binding site alterations and their DNA methylation with or without hypoxia in all three cell lines. This approach can also address the key question from their Nature paper1 - does DNA hypermethylation affect the binding of HIF1?
3. In experiments concerning 5-aza-2'-deoxycytidine (DAC) treatment, the condition was 1 μ M DAC for 3 days. Under such high concentration for 3 days, it will be very difficult to have any survival cells in 5-7 days because of high toxicity. The major discovery of DNA methylation inhibitors for cancer therapy led by Drs. Jones, Baylin, and Issa, such as 5-aza-2'-deoxycytidine (DAC) and 5-aza-2'-oxycytidine (AZA), is a low dose treatment to achieve DNA demethylation and anti-tumor memory without introducing toxicity. The authors should check the conditions of DAC treatment in vitro, such as ref. 31 or Muvarak et al, Cancer Cell, 2016, and re-do these experiments.

Reference:

- 1 Thienpont, B. et al. Tumour hypoxia causes DNA hypermethylation by reducing TET activity. Nature 537, 63-68, doi:10.1038/nature19081 (2016).
- 2 Muvarak, N. E. et al. Enhancing the Cytotoxic Effects of PARP Inhibitors with DNA Demethylating Agents - A Potential Therapy for Cancer. Cancer cell 30, 637-650, doi:10.1016/j.ccell.2016.09.002 (2016).

Reviewer 2

Are you able to assess all statistics in the manuscript, including the appropriateness of statistical tests used? Yes, and I have assessed the statistics in my report.

Comments to author:

This is an ambitious, well-executed, and well-written manuscript in which the authors use an incredible range approaches to dissect the intricate manner in which DNA methylation status, induction of gene expression by hypoxia via the HIF transcription factor dimers, and cryptic bi-directional expression of genomic repeats fit together. A wide variety of approaches is combined in a sensible manner, always with one analysis naturally leading to the next. Not only does the manuscript provide rich insight into the underlying molecular mechanisms, it also makes an exciting suggestion that DNA methylation inhibitors can push cancer cells to a state that is more susceptible to immunotherapy.

What makes the manuscript especially appealing is that in addition to using many different assays (in vitro measurement of binding constants, ChiP-seq/BS-seq, RNA-seq, including scRNA-seq, analysis of large cancer compendia, tumor graft in mouse models) a wide variety of perturbations (hypoxia, pharmacological perturbations of methylation state, CRISPR knockout of TF, demethylase and methyltransferase mutants, different cell lines, genomic integration of DNA fragment) is used to break the confounding that often plagues the questions addressed here. The authors do not shy away from reanalyzing the raw data when building on existing resources, and even developed a new software tool for dealing with technical challenges associated with interpreting repeat regions properly.

I have only a few detailed comments in addition to the above:

- page 9, top paragraph (lines 221-235): This paragraph was the only one where the conclusion at the end did not seem to be supported by the text preceding it. Please revise for clarity, so that it more obvious how the "methylation differences" come into the story here.
- page 12, line 319: evolutionary => evolutionarily
- page 14, line 389: CpGs in cryptic ("in" missing)
- page 15, line 403: the second P-value could be not significant simply because the second sample is smaller; can the authors explicitly provide sample size here, and here and elsewhere specify the test that was used to generate the P-value?
- page 17, line 459: typo in "CRISPR"
- page 17, line 465-6: "much slower" => much more slowly

Reviewer #1: This manuscript by Flora D'Anna et al and her colleagues has demonstrated that DNA methylation blocks the binding of hypoxia-inducible transcription factors (HIFs), and that DNA demethylation enables HIF binding to various repetitive genomic regions, inducing the expression of cryptic transcripts in a hypoxia dependent manner, thus further sensitizing tumors to checkpoint immunotherapy. This study tries to build a linkage between hypoxia, DNA hypomethylation, and tumor immunotolerance by providing experimental approaches, however, the data seems to conflict with their original novel discovery (Thienpont et al. Nature, 2016).

Here are some major concerns regarding this study:

1. As mentioned before, their previous study (Thienpont et al. Nature, 2016) revealed a novel discovery in the field that tumor hypoxia caused DNA hypermethylation by reducing TET activity and was associated with increased malignancy, poor prognosis and resistance to radio- and chemotherapy. However, in this current study, the authors showed that hypoxia could induce HIF binding to the hypomethylated DNA without comparing hypermethylated sites by HIF under hypoxia situation. Therefore, under hypoxia conditions, HIF binding sites should be decreased because of DNA hypermethylation. What is the consequence of the lost HIF binding, and does this increase or decrease immunotolerance? The authors need to present data and discuss the logical approach in this study.

We thank the reviewer for raising this important point. Both papers are not in conflict with one another, and we have now clearly explained this in the revised manuscript. Specifically, our previous findings published by Thienpont *et al.* (Nature 2016)¹ are independent from the current findings (D'Anna *et al.*) for 3 important reasons:

- Firstly, the TET- and HIF-dependent mechanisms are molecularly independent of one another. Indeed, the publication by Thienpont *et al.* related specifically to oxygen shortage, directly causing reduced TET activity and DNA hypermethylation. We demonstrated that oxygen acts as a direct co-factor of the TET enzymes and hence, that the DNA hypermethylation effect was independent of HIF activation (*cf.* Figure 1G of Thienpont *et al.*). Likewise, the DNA methylation-dependent HIF binding effects we observe in D'Anna *et al.* are independent of TET activity. This, we demonstrated in Figure 2B of the current manuscript using TET triple-KO cell lines.

- Secondly, as TET is mainly active in promoters and enhancers, the DNA hypermethylation effects induced by hypoxia were also confined to these regions (*cf.* Figure 2B of Thienpont *et al.*, Nature 2016). They were not observed at repressed and heterochromatic regions which therefore maintain high methylation levels during hypoxia. Tumors, however, are well-established to have reduced DNA methylation levels in heterochromatic regions, an effect that has been proposed to underlie their genomic instability. We observed that, as a result, hypomethylation in tumours specifically in heterochromatic regions enables HIF binding leading to cryptic transcript expression (and increased immunogenicity). Our current manuscript does not deal with the mechanisms underlying the reduced DNA methylation of tumors in heterochromatic regions, neither do we propose hypoxia as a potential driver of this hypomethylation.

- Finally, hypermethylation at promoters and enhancers described by Thienpont *et al.*¹ appeared as a result of prolonged hypoxia and epigenetic selection, with no significant changes being detectable after acute hypoxia. On the contrary, the hypoxic upregulation of HIF target genes and cryptic transcripts observed in D'Anna *et al.* results from an acute effect of hypoxia (16 hours of hypoxia). Importantly, such acute hypoxia does not affect the methylation status of the cells, consistent with the concept that only prolonged hypoxia induces this hypermethylation. For instance, in Thienpont *et al.*, effects of hypoxia (0.5% O₂) on DNA hypermethylation in MCF7 cells were undetectable 24 hours after hypoxia.¹ Again, the effects of (acute) hypoxia on cryptic transcripts mostly stem from repressed and

heterochromatic regions normally showing high methylation levels. Both mechanisms are hence not in conflict with one another.

To address this concern of the reviewer, we have now rephrased the introduction (line 80): *"We recently demonstrated that pervasive and ablating conditions of tumour hypoxia drive DNA methylation of tumour suppressor genes by reducing the activity of TET DNA demethylases¹. An outstanding question is, however, if and how DNA methylation in turn also influences the response of tumours to (acute) hypoxia."*

and we wrote at the start of the results section (line 125): *"To investigate the role of DNA methylation in HIF binding, we stabilized HIFs in MCF7 breast cancer cells by culturing them under acute hypoxia (0.5% O₂ for 16 hours; Figures S1a and S2, and Methods), conditions we have previously shown to be insufficient to drive hypoxia-induced hypermethylation."*

We also now highlighted this notion more explicitly in the discussion (line 515) *"We note that we did not model chronic but only acute hypoxia in vitro, conditions that do not directly alter DNA methylation and that are thus distinct from the prolonged, chronic hypoxia we previously described to be essential to cause DNA hypermethylation by TET inhibition at promoters and enhancers¹."*

Finally, in response to question 2, we have also demonstrated that this acute hypoxia does not alter DNA methylation, as detailed below.

2. Following question 1, the experimental approach (Figure 1) is misleading, as the authors try to demonstrate the correlation between HIF1 binding sites and DNA methylation status using normoxic conditions. The best way to answer the question regarding alterations of HIF binding and DNA methylation under hypoxia, the authors should compare HIF binding site alterations and their DNA methylation with or without hypoxia in all three cell lines. This approach can also address the key question from their Nature paper¹ - does DNA hypermethylation affect the binding of HIF1?

We previously did not test whether HIF binding is dependent on the DNA methylation level of its binding sites following acute hypoxia, as we knew that acute hypoxia has no effects on DNA methylation. We apologize if this comes across as misleading, and have detailed this rationale more clearly in the revised manuscript (*cf.* response to question 1).

Moreover, as requested, we have now assessed HIF1 β binding under both normoxic (21% O₂) and hypoxic conditions (16 hours of 0.5% O₂; hypoxia) and this for the 3 cell lines that were profiled in the original manuscript (*i.e.*, RCC4, MCF7 and SK-MEL-28). We compared the methylation status of the Hypoxia Response Element (HRE) for the shared (n=6,152) and cell type-specific (n=7140) HIF1 β binding sites. As expected, we did not see a difference in the methylation status between normoxic and hypoxic conditions: shared binding sites bound by HIF1 β were unmethylated in all 3 cell lines, both under normoxic and hypoxic conditions, while cell type-unique binding sites were unmethylated only in the cell line where there was HIF1 β binding, both under normoxic and hypoxic conditions. These data thus clearly demonstrate that under acute hypoxia conditions, there is no DNA hypermethylation affecting HIF binding. These new data were added to the revised manuscript in Figure 1e-f and on line 168.

Figure 1: (a) Heatmaps of HIF1 β binding under hypoxia (red) and of DNA methylation as estimated using SeqCapEpi BS-seq (blue) at regions flanking the HIF1 β ChIP-seq peak summit (\pm 5 kb). (top) HIF1 β binding peaks shared between the 3 cell lines. (bottom) HIF1 β binding peaks unique to each cell line. Heatmaps depict RPKM of HIF1 β ChIP-seq and % DNA methylation. HIF1 β binding was assessed after 16 hours of 0.5% O₂ (hypoxia) and DNA methylation under 21% O₂ (normoxia) or after 16 hours of 0.5% O₂ (hypoxia). (b-c) Quantification of the DNA methylation level at HIF1 β binding peak summits \pm 100 bps, for peaks that are shared between or unique to one of the 3 cell lines, for cells grown either under 21% O₂ (normoxia, b) or after 16 hours of 0.5% O₂ (hypoxia, c). This figure was added to Figure 1e-f.

3. In experiments concerning 5-aza-2'-deoxycytidine (DAC) treatment, the condition was 1 μ M DAC for 3 days. Under such high concentration for 3 days, it will be very difficult to have any survival cells in 5-7 days because of high toxicity. The major discovery of DNA methylation inhibitors for cancer therapy led by Drs. Jones, Baylin, and Issa, such as 5-aza-2'-deoxycytidine (DAC) and 5-aza-2'-oxycytidine (AZA), is a low dose treatment to achieve DNA demethylation and anti-tumor memory without introducing toxicity. The authors should check the conditions of DAC treatment in vitro, such as ref. 31 or Muvarak et al, Cancer Cell, 2016, and re-do these experiments.

We thank the reviewer for bringing up this important issue. We note that effects of 5-aza-2'-deoxycytidine (AZA) are highly cell-type-dependent, as shown for example by Zong and colleagues for 21 gastric cancer cell lines.² The DNA demethylating effects for example depend on the number of cell divisions that take place during the exposure. As such, more slowly dividing cells can survive longer than fast-growing cells. In this manuscript, we performed the majority of the experiments in MCF7 cells, which unfortunately were not included in Muvarak

et al. (Cancer Cell 2016)³. However, Beyrouthy *et al.* did already describe that MCF7 cells, together with osteosarcoma (U2OS), and colon (HCT116) cancer cells are relatively resistant to AZA, and unaffected in viability by the concentrations (1 μ M) utilized here.⁴ Likewise, for the majority of our experiments, MCF7 cells were pre-treated with AZA (1 μ M) for 3 days, which was followed by another day of exposure to AZA in hypoxia or normoxia, bringing the total AZA exposure time for experiments to 4 days.

To further alleviate the reviewer's concern and provide direct evidence that aza is functional but not cytotoxic in MCF7 cells under the conditions of our experiments, we used sulforhodamine B (SRB) assays to quantify potential cytotoxicity, and LC/MS to quantify the effects of AZA on DNA methylation levels. Briefly, following cell seeding and attachment, cells were exposed for 96h to various concentrations of 5-aza-2'-deoxycytidine, prior to fixation and protein quantification using sulforhodamine B. Importantly, this revealed no cell death at 1 μ M of AZA, in line with the earlier reports described higher. At the 5mC level, we observed a reduction in DNA methylation of \sim 60% at this dose, and no significant reduction at lower doses (Figure 2). Overall, these data demonstrate that the conditions utilized here are state-of-the-art, necessary and sufficient to reduce DNA methylation levels in MCF7 cells, and not associated with a significant cytotoxicity.

Figure 2: Effects of exposure to different concentrations of 5-aza-2'-deoxycytidine (AZA) on cell proliferation as measured by sulforhodamine B assays (left), and on DNA methylation (5mC/C) as measured by LC/MS (right). Added to Figure S6f.

We therefore altered line 314 as follows: "We then assessed whether a similar phenomenon is at play in cancer cell lines, and pharmacologically demethylated MCF7 cells using a non-cytotoxic⁴ dose of 5-aza-2'-deoxycytidine (aza, 1 μ M), necessary and sufficient for strongly reducing DNA methylation (Figure S6f)."

References:

1. Thienpont, B. *et al.* Tumour hypoxia causes DNA hypermethylation by reducing TET activity. *Nature* **537**, 63-68 (2016).
2. Zong, L. *et al.* LINC00162 confers sensitivity to 5-Aza-2'-deoxycytidine via modulation of an RNA splicing protein, HNRNPH1. *Oncogene* **38**, 5281-5293 (2019).
3. Muvarak, N.E. *et al.* Enhancing the Cytotoxic Effects of PARP Inhibitors with DNA Demethylating Agents - A Potential Therapy for Cancer. *Cancer Cell* **30**, 637-650 (2016).
4. Beyrouthy, M.J. *et al.* High DNA methyltransferase 3B expression mediates 5-aza-deoxycytidine hypersensitivity in testicular germ cell tumors. *Cancer Res* **69**, 9360-6 (2009).

Reviewer #2: This is an ambitious, well-executed, and well-written manuscript in which the authors use an incredible range approaches to dissect the intricate manner in which DNA methylation status, induction of gene expression by hypoxia via the HIF transcription factor dimers, and cryptic bi-directional expression of genomic repeats fit together. A wide variety of approaches is combined in a sensible manner, always with one analysis naturally leading to the next. Not only does the manuscript provide rich insight into the underlying molecular mechanisms, it also makes an exciting suggestion that DNA methylation inhibitors can push cancer cells to a state that is more susceptible to immunotherapy.

What makes the manuscript especially appealing is that in addition to using many different assays (in vitro measurement of binding constants, ChiP-seq/BS-seq, RNA-seq, including scRNA-seq, analysis of large cancer compendia, tumor graft in mouse models) a wide variety of perturbations (hypoxia, pharmacological perturbations of methylation state, CRISPR knockout of TF, demethylase and methyltransferase mutants, different cell lines, genomic integration of DNA fragment) is used to break the confounding that often plagues the questions addressed here. The authors do not shy away from reanalyzing the raw data when building on existing resources, and even developed a new software tool for dealing with technical challenges associated with interpreting repeat regions properly.

I have only a few detailed comments in addition to the above:

- page 9, top paragraph (lines 221-235): This paragraph was the only one where the conclusion at the end did not seem to be supported by the text preceding it. Please revise for clarity, so that it more obvious how the "methylation differences" come into the story here.

Thank you for pointing this out. This should indeed have been phrased more clearly. The paragraph has now been revised as follows:

Start of paragraph (line 225): *"Comparison of our 7,153 HIF1 β peaks to previously published HIF1 α and HIF2 α ChIP-seq data in MCF7 cells revealed that the methylation status of HIF1 β binding peaks was independent of the HIF α binding partner, as HIF1 α - and HIF2 α -bound DNA showed similar methylation levels (Figure S3h)."*

End of paragraph (line 237): *"In conclusion, the binding preferences of HIF1 α and HIF2 α differ, with HIF1 α being somewhat more promoter-enriched and HIF2 α being more enhancer-enriched, but these preferences are not determined by differences in DNA methylation at their binding sites."*

- page 12, line 319: evolutionary => evolutionarily
This was adapted accordingly (now line 325)

- page 14, line 389: CpGs in cryptic ("in" missing)
This was adapted accordingly (now line 395)

- page 15, line 403: the second P-value could be not significant simply because the second sample is smaller; can the authors explicitly provide sample size here, and here and elsewhere specify the test that was used to generate the P-value?

Thank you for pointing out this potential weakness in our analyses. The first P-value was calculated on 2,505 samples and the second on 2,681 samples. Differences are hence not due to a difference in power. Sample numbers were already detailed in the corresponding figures, and are now mentioned explicitly in the main text. We originally did not mention the test generating this P-value in the main text, but we referred to the corresponding methods section (writing "See Methods"). There, we detailed the underlying analysis as follows: *"To test the interaction between hypoxia and DNA methylation, we assessed read counts for cryptic transcripts in two negative-binomial generalized linear models with both oxygenation (hypoxic*

and normoxic; encoded as 0 and 1) and methylation (low and high methylation; encoded as 0 or 1), with or without an interaction term. Both models were compared to each other using DESeq. A positive interaction coefficient represents a cooperative enhancement of cryptic transcript expression in low-methylation, hypoxic tumours."

In order to accommodate this comment, we now revised the text as follows (line 408): "*In TCGA, this interaction was only detected in tumour types known to respond to immunotherapy (P=0.0031 in n=2,505 responsive tumours versus P=0.69 in n=2,681 non-responsive tumours; see Methods for a detailed description of the generalized linear model; Figure 5b-c).*"

At all other instances, statistical tests are detailed in the figure legends where appropriate, or in the main text if we are not referring to a figure.

- page 17, line 459: typo in "CRISPR"
This was adapted accordingly (now line 465)

- page 17, line 465-6: "much slower" => much more slowly
This was adapted accordingly (now line 471)

Second round of review

Reviewer 1

The authors have answered most my concerns. The quality of this paper has been dramatically improved.